



# Pronounced seasonal and spatial variability in determinants of phytoplankton biomass dynamics along a near–offshore gradient in the southern North Sea

Viviana Otero[1,§], Steven Pint[1,§], Klaas Deneudt[1], Maarten De Rijcke[1], Jonas Mortelmans[1], Lennert Schepers[1], Patricia Cabrera[1], Koen Sabbe[2], Wim Vyverman[2], Michiel Vandegehuchte[1] and Gert Everaert[1]

1 Flanders Marine Institute, Wandelaarkaai 7, Ostend, 8400, Belgium
2 Department of Biology, Ghent University, Krijgslaan 281-S8, Ghent, 9000, Belgium
§These authors contributed equally to this work

*Correspondence to:* Steven Pint (steven.pint@vliz.be)

## Abstract

Marine phytoplankton biomass dynamics are affected by eutrophication, ocean warming, and ocean acidification. These changing abiotic conditions may impact phytoplankton biomass and its spatiotemporal dynamics. In this study, we used a nutrient–phytoplankton–zooplankton model to quantify the relative importance of bottom-up and top-down determinants on phytoplankton biomass dynamics in the Belgian Part of the North Sea. Using four years (2014 – 2017) of monthly observations at nine locations of nutrients, solar irradiance, sea surface temperature, chlorophyll-a and zooplankton biomass, we disentangled the monthly, seasonal and yearly variation in phytoplankton biomass dynamics. To quantify how the relative importance of determinants changed along a near–offshore gradient, the analysis was performed for three spatial regions, i.e. nearshore region (< 10 km to the coastline), midshore region (10 – 30 km), and offshore region (> 30 km). We found that from year 2014 to 2017, phytoplankton biomass dynamics ranged from 1.4 to 23.1 mg Chla m$^{-3}$. Phytoplankton biomass dynamics follow a general seasonal cycle as in other temperate regional seas, with a distinct spring bloom (5.3 – 23.1 mg Chla m$^{-3}$) and a modest autumn bloom (2.9 – 5.4 mg Chla m$^{-3}$). This seasonal pattern was most expressed in the nearshore region. The relative contribution of factors determining phytoplankton biomass dynamics varied spatially and temporally. Throughout a calendar year, solar irradiance and zooplankton grazing were the most influential determinants in all regions, i.e. explained 38% – 65% of the variation in the offshore region, 45% – 71% in the midshore region, and 56% – 77% in the nearshore region. In the near- and midshore regions, nutrients are most limiting the phytoplankton production in the month following the spring bloom (44% – 55%). Nutrients are a determinant throughout the year in the offshore region (27% – 62%). During winter, sea surface temperature is a determinant in all regions (15% – 17%). The findings of this study contribute to a better mechanistic understanding of the spatiotemporal dynamics of phytoplankton biomass in the southern North Sea. The parameterized causal relationships allow estimating how the base of the southern North Sea food web will change under future climate change and/or blue economy activities that affect one or more determinants of the phytoplankton biomass dynamics.





**Keywords:**

Primary production, Ecosystem model, Phytoplankton biomass dynamics, Environmental conditions

**Introduction**

Marine phytoplankton, which forms the base of marine food webs, is responsible for about half of the world's total primary

production (Field et al., 1998). Net global marine primary production is estimated at 50.7 Gt carbon per annum (Carr et al., 2006). Principal factors that determine marine primary production are solar irradiance, nutrient availability, and sea surface temperature (SST), mainly by limiting its growth rate and carrying capacity. According to Liebig's law of the minimum, phytoplankton production will be as high as allowed by the least available resource (de Baar, 1994). However, in many cases co-limitation by resources is a better description of the factors that influence phytoplankton biomass dynamics (Harpole et al.,

2011; Price and Morel, 1991). Living in the Anthropocene, phytoplankton biomass dynamics are affected by human activities that directly or indirectly alter the abiotic marine environment, such as the burning of fossil fuels, eutrophication, and chemical pollution. To date, we have limited insight into how the combination of changing conditions may affect phytoplankton biomass dynamics at high-resolution spatiotemporal scales.

In temperate marine regions, phytoplankton biomass dynamics follows an annual cycle consisting of spring and autumn phytoplankton blooms followed by periods of zooplankton grazing. Phytoplankton blooms are triggered by high nutrient availability and sufficient solar irradiance (Irigoien et al., 2005). After a few weeks of rapid growth, phytoplankton biomass becomes restricted by nutrient limitation and zooplankton grazing. As in other parts of the North Sea, the most important factors that determine the phytoplankton biomass in the Belgian part of the North Sea (BPNS) are nutrient concentrations, SST

and solar irradiance (Arndt et al., 2011; Blauw et al., 2018; Capuzzo et al., 2018; Desmit et al., 2020). Everaert et al. (2015) is one of the first studies that quantified the relative importance of these conditions in the BPNS. Based on a relatively short time series at one location at the BPNS, it was found that SST and solar irradiance accounted for 20% (summer) to 50% (winter) of the observed seasonal variation (Everaert et al., 2015) and can thus be considered potentially key determinants. Nutrients appeared to be less determining than SST and solar irradiance in the BPNS (Everaert et al., 2015), which was also found for

the entire North Sea by Llope et al. (2009) and McQuatters-Gollop et al. (2007). Nutrients become the dominant determinant of phytoplankton biomass (30%) in the month after the phytoplankton bloom. Besides these bottom-up determinants, there is also a strong top-down control of the phytoplankton biomass dynamics by zooplankton grazing, i.e. up to 50% of the phytoplankton growth limitation (Everaert et al., 2015). However, the BPNS is a heterogeneous and highly dynamic coastal area, so it is doubtful whether the quantifications in Everaert et al. (2015) are generalizable for the entire BPNS. The BPNS is

relatively shallow with water depths gradually increasing to 45 m from the southeast towards the northwest (Van Lancker et al., 2015). Sea surface temperatures vary seasonally between 5°C and 20°C. The salinity is strongly influenced by the river plumes of the Scheldt, Rhine, Seine and Meuse (Lacroix et al., 2004) and varies between 29 to 35 PSU. Seawater from the

English Channel, which contains run-off from the Seine, flows northwards through the BPNS driven by the anti-clockwise current in the North Sea (Turrell, 1992). In case of nutrients, the Seine plays a major role, except in the vicinity of the Scheldt estuary and in the northern part of the BPNS i.e. mainly influenced by the influx of water of the Atlantic Ocean (Lacroix et al., 2007). Overall, in heterogeneous and dynamic coastal areas such as the BPNS, the relative contribution of bottom-up and top-down determinants may be likewise dynamic, both spatially and temporally. The BPNS being a prime example of such a system in combination with the availability of long term observations with high spatial resolution offers unique possibilities to study the scales at which the relative contributions of bottom-up and top-down determinants may shift. Having a better understanding of the spatial variation in the relative contribution of the determinants of phytoplankton biomass dynamics, can lead to a more adjusted and specified management of the BPNS and the further development of the blue economy in Belgium.

In the present research, the aim is to analyse which factors drive marine phytoplankton biomass dynamics in the BPNS and how their relationship to primary production varies on a spatiotemporal scale. In particular, we analysed how the relative contributions of SST, nutrient regimes, solar irradiance and zooplankton grazing to the marine phytoplankton biomass change spatially and temporally. We used a nutrient–phytoplankton–zooplankton model from Soetaert and Herman (2009) adjusted by Everaert et al. (2015) to simulate changes in plankton density in the nearshore, midshore and offshore regions, based on monthly data collected from 2014 to 2017 at ten sampling locations in the BPNS.

**Materials and Methods**

**2.1 Input data**

Three regions of interest were studied (Fig. 1), i.e. the nearshore region (10 km), the midshore region (10 – 30 km) and the offshore region (> 30 km). These regions were defined based on an integration of information about their distance to the coast, sediment composition, bathymetry (Ivanov et al., 2021; Maes et al., 2020), and prior knowledge about different abiotic conditions (Ivanov et al., 2020). LifeWatch measuring stations are located in each of these regions (indicated as triangles in Fig. 1). LifeWatch is a European Research Infrastructure within the European Strategy Forum on Research (ESFRI) that focuses on biodiversity research and activities, such as measuring of the biotic and abiotic environment. LifeWatch stations in the near- and midshore are visited monthly (Mortelmans et al., 2019). We used data from two stations, i.e. 130 for nearshore and 330 for midshore and pooled the data of seven sampling stations for the offshore region (Fig. 1). The data of all offshore stations, i.e. seven stations, were pooled in this study due to the lower temporal sampling. In the offshore region, the LifeWatch stations are visited seasonally, whereas stations in the near- and midshore are visited monthly.

Four open-access datasets obtained from LifeWatch sampling campaigns, which are coordinated by the Flanders Marine Institute, at nine locations in the BPNS from 2014 to 2017 were used. A first open-access dataset related to the nutrient concentrations, i.e. $NH_4$, $NO_3$, $NO_2$, $PO_4$ and $SiO_4$, was used and the measurements were performed by a Skalar AutoAnalyser





system (VLIZ, 2021a; Mortelmans et al., 2021). A second open-access dataset consisted of seawater temperature measurements performed by a CTD (VLIZ, 2021a). A third open-access dataset comprised zooplankton abundances obtained through ZooScan analysis (VLIZ, 2021c; Mortelmans et al., 2021). A fourth and final dataset used in this research contained information on the in situ pigment concentrations, i.e. Chla the measurements were performed by HPLC (VLIZ, 2021a Mortelmans et al., 2021).

The gathered data was assembled in two data sets, a first containing the data related to nutrient and pigment concentrations, and seawater temperature, and a second for the zooplankton abundances. Only data of the selected taxa, i.e. Calanoida, Noctiluca, Harpacticoida and Appendicularia, were used in this study. All data were timestamped and were location specific.

  In addition to the SST dataset gathered from LifeWatch, a second SST dataset with a higher temporal coverage was required

to infer daily time series, i.e. input data for our nutrient–phytoplankton–zooplankton model, by means of generalized additive models. SST data from the Westhinder station, i.e. data from the Westhinder measuring pile complemented with data from Westhinder - buoy (2% of the data set), that is part of the Flemish Banks Monitoring Network (MVB; IVA MDK, n.d.) was used to infer daily time series for nutrients and SST. Due to large data gaps in the SST data set of the Westhinder station in 2018, we have chosen to use the data set from 2014 – 2017 in order to avoid increasing random noise when calibrating the

model by including an extra data set of SST.

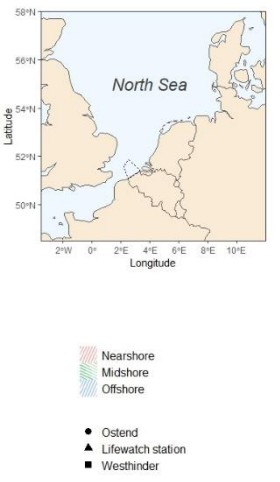
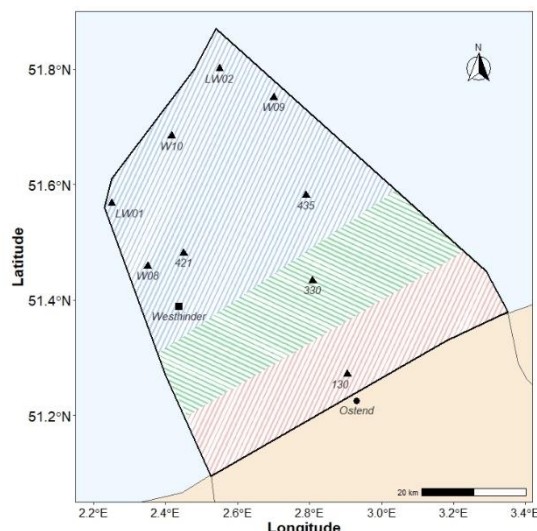

**Figure 1. Map of the Belgian part of the North Sea (BPNS) showing the sampling locations used in this study for the near-, mid- and offshore regions. The square mark shows the Westhinder station from the Flemish Banks Monitoring Network in the BPNS. The**
**black outline indicates the Belgian Exclusive Economic Zone.**

## 2.2 Time trends for input data

Generalized additive models (GAM) were used to infer daily trends for nutrient concentrations, i.e. N, P and Si, in the three regions of interest (Appendix D). The daily time trends are used as input for the nutrient–phytoplankton–zooplankton model.





Dissolved inorganic nitrogen (DIN) was calculated as the sum of ammonium ($NH_4^+$), nitrate ($NO_3^-$) and nitrite ($NO_2^-$).

Measurements of $NH_4$, $NO_3$, $NO_2$, $PO_4$ and $SiO_4$ registered in the LifeWatch database were converted into nitrogen, phosphorus and silica equivalent weight, respectively. The covariates used in the GAM models were month and year as in Everaert et al. (2015). The Akaike Information Criterion (AIC) was calculated using the package 'stats' (R Core Team, 2018) and the minimum AIC was used to select the best-fit GAM models (Table E2). The corresponding $R^2$ of the best-fit GAM model provides an indication of how well the model fits the observational data (Table E1).

Daily SST for near- and midshore stations were calculated using GAM models based on temperature observations from LifeWatch. For the offshore region, SST data of the Westhinder station (Fig. 1) were used. In case that multiple temperature loggings were available per day, the median daily SST was used as input for the model.

## 2.3 Ecological model

A nutrient–phytoplankton–zooplankton (NPZ) model as presented in Everaert et al. (2015), which was adjusted from Soetaert

and Herman (2009), was used to simulate spatiotemporal changes in plankton density in the BPNS from 2014 to 2017. The model time step is expressed in mmol N $m^{-3}$ $d^{-1}$ and the state variables are expressed in mmol N $m^{-3}$. The necessary input data for the NPZ model are (i) nutrient concentrations, (ii) SST, and (iii) solar irradiance (photosynthetically active radiation (PAR)) corrected for the diffusion attenuation coefficient ($K_d$; Fig. 2, Appendix A). PAR and nutrients, i.e. DIN, $PO_4$ and $SiO_4$, were implemented in the NPZ model as saturating Michaelis–Menten equations as in Arndt et al. (2011). As such, the Michaelis–

Menten equations describe the determining factor of a variable of interest, i.e. PAR, DIN, $PO_4$ or $SiO_4$, for each time step. The outcome of these equations varies from one to zero, with a value of one indicating no limitation for plankton growth, and a value of zero indicating a complete limitation of the plankton growth (Soetaert and Herman, 2009). The amount of PAR available was corrected for the diffuse attenuation coefficient by means of the Lambert–Beer law (Kirk, 1994; Lund-Hansen, 2004). The influence of SST on the plankton growth followed the Thomann and Mueller (1987) equation (Appendix A). A

detailed equation-based description of the model and its variable-specific forcing functions is available in Appendix A. The NPZ model and corresponding calculations were performed in R (R Core Team, 2018; version 3.4.4; R packages in Appendix A).





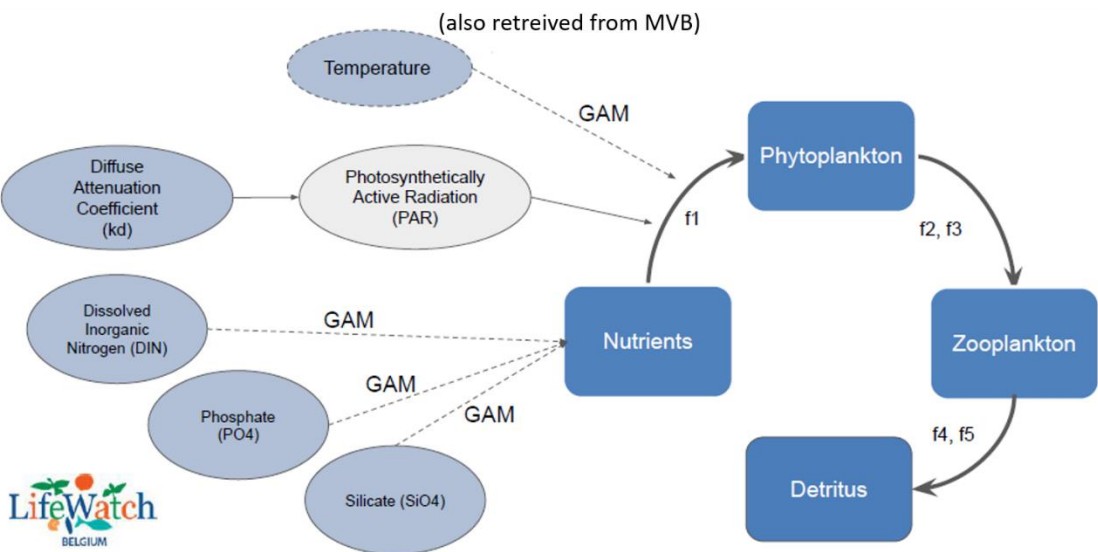

**Figure 2. Structure of the nutrient–phytoplankton–zooplankton ecological model. Input data were obtained from LifeWatch and**
**Flemish Banks Monitoring Network (MVB) regarding sea surface temperature. Generalized additive models (GAM) were used to**
**obtain daily data for nutrients based on monthly observations (see section 2.1).**

### 2.4 Selection of model parameters and validation

The NPZ model requires parameterization of thirteen parameters (Table 1 and B1). For the model parameterization, we
followed a two-step approach to find the most optimal model calibration for each region of the BPNS to mimic the
biogeochemical processes in each respective region (Fig. 3). In a first step, 5,000 unique sets of parameters were run for each
region of interest. The initial minimum and maximum parameter values used to define these unique sets of parameters were
based on values reported in literature (Appendix B). In a second step, a new set of 5,000 unique parameterizations were run
for each region of interest and for two seasons, i.e. spring and autumn. In this second step, the minimum and maximum
parameter values were based on the 10% best parameterization for either spring or autumn conditions from the first step. To
rank these 10% best models, we calculated the Root Mean Square Error (RMSE) by comparing the NPZ model predictions of
phyto- and zooplankton density with the observed pigment chlorophyll-a (VLIZ, 2021a), i.e. a proxy for phytoplankton
biomass, and zooplankton densities (VLIZ, 2021c), respectively. We calculated the total RMSE for each unique
parameterization as the cumulated error for phytoplankton and zooplankton. For each unique model parameterization that we
tested, we cumulated the error over the different time steps. We considered the best parameterizations as those with the lowest
10% RMSE. In this second step, we retained a set of parameters (Table 1) to describe spring conditions, i.e. the situation after
winter solstice and before summer solstice, and a set of parameters to describe the autumn conditions, i.e. situation after
summer solstice and before winter solstice.



**Figure 3. Key steps of model development, i.e. calibration, fitting and validation. In a first step, 5,000 unique sets of parameters were**
**run for each region of interest. The initial minimum and maximum parameter values used to define these unique sets of parameters**
**were based on values reported in literature (Appendix B). The nutrient–phytoplankton–zooplankton (NPZ) model is driven by daily**
**nutrient and sea surface temperature data generated from LifeWatch and Flemish Banks Monitoring Network (MVB) observations.**
**In a second step, a new set of 5,000 unique parameterizations were run for each region of interest and for two seasons, i.e. spring**
**and autumn. In this second step, the minimum and maximum parameter values were based on the 10% best parameterization for**
**either spring or autumn conditions from the first step. To rank these 10% best models, we calculated the Root Mean Square Error**
**(RMSE) by comparing the NPZ model predictions of phyto- and zooplankton density with the observed pigment chlorophyll-a and**
**zooplankton densities, respectively.**

To assess the model fit, we compared the NPZ phytoplankton biomass predictions (expressed in mmol N m$^{-3}$) with the observed

phytoplankton biomass data (expressed in mg Chla m$^{-3}$). To enable the comparison, we converted the unit of the NPZ

phytoplankton biomass predictions and the unit of the LifeWatch phytoplankton biomass data to mg Chla m$^{-3}$. To do so, we



used the Chl:N ratio parameter of the corresponding simulation (Table 1). To compare the NPZ zooplankton production predictions (expressed in mmol N m$^{-3}$) with the LifeWatch zooplankton observations (expressed in ind m$^{-3}$), we converted the latter to the same unit. To do so, we calculated the taxon-specific body mass per individual (mg C ind$^{-1}$), converted this mass to a molar mass (mmol C m$^{-3}$), and used a taxon-specific C:N ratio to convert the molar mass finally to mmol N m$^{-3}$. Details
about this conversion are available in Appendix C (Table C1 & C2).

**Table 1. The seasonal minimum and maximum values of the thirteen parameters used as input for the nutrient–phytoplankton–zooplankton model for each region.**

| Parameter | Unit | Period | Nearshore region | Midshore region | Offshore region |
|---|---|---|---|---|---|
| maxUptake | day$^{-1}$ | Spring | 0.38 – 0.66 | 0.38 – 0.61 | 0.50 – 1.12 |
| | | Autumn | 0.40 – 0.78 | 0.38 – 0.80 | 0.38 – 0.90 |
| excretionRate | day$^{-1}$ | Spring | 0.16 – 0.18 | 0.12 – 0.16 | 0.11 – 0.15 |
| | | Autumn | 0.11 – 0.17 | 0.11 – 0.15 | 0.11 – 0.14 |
| maxGrazing | day$^{-1}$ | Spring | 0.87 – 0.96 | 0.85 – 0.92 | 0.88 – 0.97 |
| | | Autumn | 0.88 – 0.96 | 0.85 – 0.93 | 0.89 – 0.97 |
| ksGrazing | mmol N m$^{-3}$ | Spring | 2.15 – 3.27 | 1.54 – 2.15 | 1.48 – 2.22 |
| | | Autumn | 1.31 – 2.27 | 1.19 – 1.59 | 1.25 – 1.94 |
| pFaeces | day$^{-1}$ | Spring | 0.29 – 0.41 | 0.27 – 0.40 | 0.27 – 0.40 |
| | | Autumn | 0.25 – 0.40 | 0.24 – 0.32 | 0.25 – 0.37 |
| mortalityRate | (mmol N m$^{-3}$)$^{-1}$ day$^{-1}$ | Spring | 0.28 – 0.39 | 0.28 – 0.41 | 0.29 – 0.41 |
| | | Autumn | 0.32 – 0.44 | 0.35 – 0.42 | 0.33 – 0.45 |
| ChlNratio | mg Chla (mmol N)$^{-1}$ | Spring | 7.00 – 7.86 | 6.78 – 7.60 | 6.65 – 7.47 |
| | | Autumn | 6.62 – 7.55 | 5.33 – 6.84 | 4.33 – 6.61 |
| ksPAR | Einst m$^{-2}$ s$^{-1}$ | Spring | 126 – 227 | 133 – 224 | 103 – 210 |
| | | Autumn | 121 – 205 | 126 – 200 | 115 – 210 |
| Tobs | °C | Spring | 9.86 – 13.84 | 10.11 – 13.29 | 9.54 – 13.62 |
| | | Autumn | 10.41 – 12.83 | 9.66 – 13.86 | 10.08 – 13.62 |
| ksDIN | mmol N m$^{-3}$ | Spring | 1.33 – 4.21 | 1.62 – 3.94 | 1.92 – 3.70 |
| | | Autumn | 1.17 – 3.64 | 2.22 – 4.11 | 2.07 – 4.29 |
| ksP | mmol P m$^{-3}$ | Spring | 0.30 – 0.43 | 0.28 – 0.44 | 0.30 – 0.44 |
| | | Autumn | 0.33 – 0.40 | 0.29 – 0.46 | 0.30 – 0.44 |
| ksSi | mmol Si m$^{-3}$ | Spring | 0.41 – 0.66 | 0.43 – 0.67 | 0.34 – 0.67 |
| | | Autumn | 0.35 – 0.63 | 0.40 – 0.67 | 0.39 – 0.65 |





| Kd* | m⁻¹ | Spring | 0.73– 0.90 | 0.44 – 0.60 | 0.28– 0.38 |
| | | Autumn | 0.77 – 0.92 | 0.45 – 0.60 | 0.28 – 0.38 |

### 2.5 Relative contributions

The relative importance of the SST, PAR, DIN, $PO_4$, $SiO_4$ and zooplankton grazing for phytoplankton biomass dynamics were calculated as in Everaert et al. (2015). To do so, we made use of the forcing functions that were integrated in the model (cfr. 2.3). For each determinant, the absolute limitation was calculated as one minus the limitation factor. Then, the relative contribution of each determinant was calculated as the absolute limitation divided by the sum of all absolute limitations. Afterwards, the monthly relative contribution of each determinant was calculated based on the average relative contribution

of the 5% best simulations, i.e. simulations with lowest RMSE, for each region of interest. Normality and homogeneity of the relative contribution data were tested by means of the Shapiro–Wilk test ($p < 0.05$) and the Levene's test ($p < 0.05$) respectively, using the packages 'stats' (R Core Team, 2018) and 'lawstat' (Gastwirth et al., 2020). Potential differences in determinants between regions were examined using the Kruskal–Wallis test ('stats' package; R Core Team, 2018) and the Dunn test ('dunn.test' package; Dinno, 2017) in R (R Core Team, 2018).

**Results**

### 3.1 Model fit

GAM models were used to create time trends of nutrients and SST (Fig. E1). SST had the highest $R^2$ values and the parameter with the lowest $R^2$ varied between regions (Table E1).

To assess the model fit, the model predictions of phyto- and zooplankton were compared to field observations, i.e. the RMSE was calculated for each unique parameterization. The nearshore region had a total RMSE of 1.24 – 1.38. The mid- and offshore regions had a total RMSE of 0.39 – 0.46 and 0.32 – 0.42, respectively. We found that the RMSE for phytoplankton in the nearshore was 1.09 – 1.31, midshore 0.30 – 0.40, and offshore 0.26 – 0.38. For zooplankton, the RMSE was 0.07 – 0.23 in the nearshore, 0.06 – 0.13 in the midshore, and 0.04 – 0.09 in the offshore.

### 3.2 Phytoplankton time trends

We found clear spatial differences in the phytoplankton biomass dynamics in the BPNS. The closer to the coastline, the higher the amplitude of the spring phytoplankton bloom and the more distinct the seasonal pattern (Fig. 4). In the nearshore region, the phytoplankton biomass ranged from 3.5 to 23.1 mg Chla m⁻³, and followed a clear seasonal trend, with highest chlorophyll-a concentration in spring and lowest chlorophyll-a concentration in winter (Fig. 4a). In the midshore region, the maximum

phytoplankton biomass was estimated to be 8.8 mg Chla m⁻³, and the minimum phytoplankton biomass was 1.8 mg Chla m⁻³ (Fig. 4b). In the offshore region, spring blooming periods were still noticeable, but less prominent (1.4 – 5.3 mg Chla m⁻³) in





terms of absolute phytoplankton biomass compared to the nearshore region. In each of the selected regions, the autumn blooming periods were noticeable, but they became relatively more pronounced with increasing distance to the coastline (Fig. 4). Overall, spring blooms were observed for each region and were followed by a smaller peak at the end of summer (Fig. 4).


We observed a response of the zooplankton population to the spring phytoplankton bloom (Fig. 5 and F2). As theoretically expected from a classic predator–prey pattern, there is a time lag between the peak in phytoplankton density and zooplankton density (Fig. 5 and F2). Note that higher, i.e. almost four times higher, zooplankton densities in the spring blooms are observed in the nearshore stations as compared to the offshore stations. In the nearshore region, zooplankton density ranged from 0.002

to 0.41 mmol N m$^{-3}$ (Fig. F1), in mid- and offshore regions these ranges were 0.003 – 0.12 mmol N m$^{-3}$ and 0.002 – 0.10 mmol N m$^{-3}$ respectively (Fig. F1).

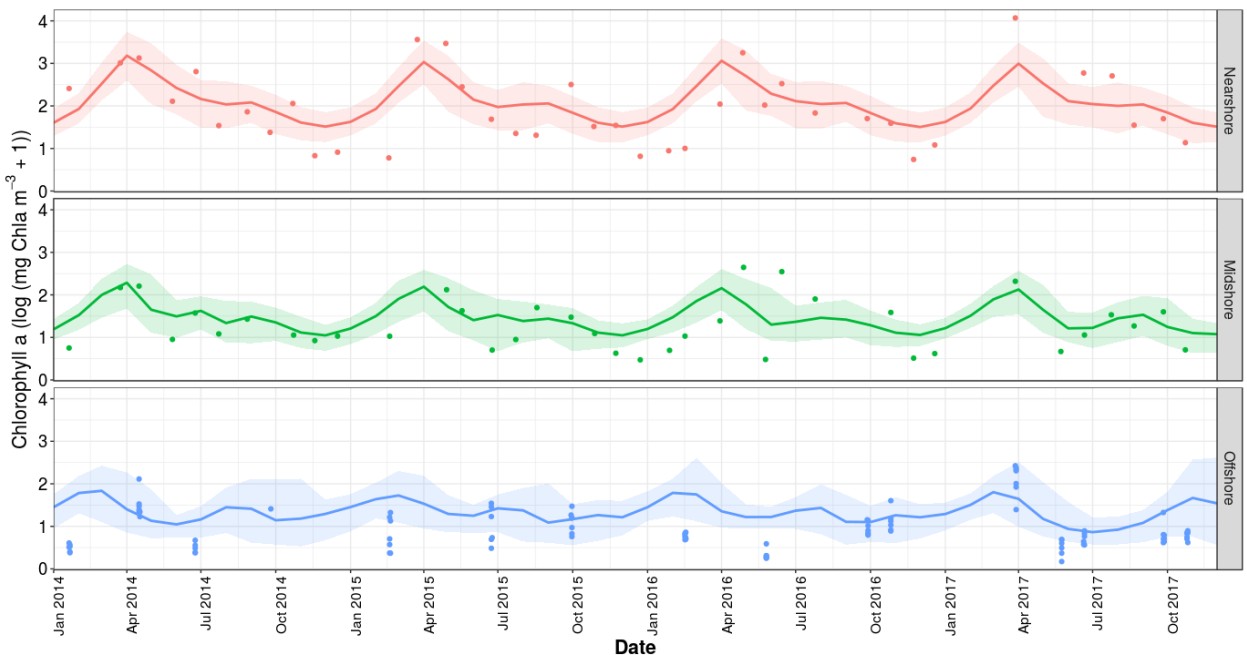

**Figure 4. Phytoplankton biomass simulations using the nutrient–phytoplankton–zooplankton model in the nearshore, midshore and**
**offshore region in the Belgian part of the North Sea. The bold lines indicate the average phytoplankton biomass predictions and the shaded areas indicate the 95% confidence interval. The dots are observed values collected during the LifeWatch campaigns.**





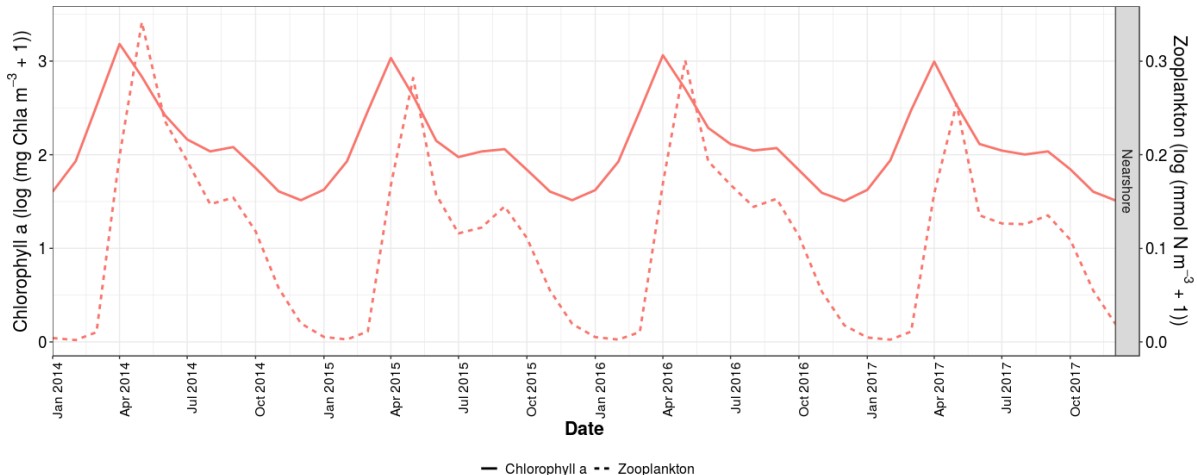

**Figure 5.** Phyto- (solid line) and zooplankton (dashed line) density simulations using the nutrient–phytoplankton–zooplankton model in the nearshore region of the BPNS. The lines represent the average phyto- and zooplankton density predictions.

### 3.3 Relative contributions

We found that solar irradiance and zooplankton grazing are the most important determinants of phytoplankton biomass throughout the year (Fig. 6). Together they contribute for 38% to 77% to the phytoplankton biomass dynamics in the BPNS. The contribution of zooplankton grazing is the highest after spring blooms (31%) and during autumn months (35%). PAR plays a more important role in phytoplankton biomass mainly in autumn (43%). We found that nutrients and SST play a relatively less important role in phytoplankton biomass dynamics. The total contribution of nutrients is maximum 44%. During spring bloom, $PO_4$ plays an important role, while $SiO_4$ and DIN take over during early summer (Fig. 6 and Appendix G). $PO_4$ is the most limiting nutrient (12% – 29%), followed by DIN (3.6% – 26%) and $SiO_4$ (1.2% – 15%), respectively. The SST only plays a limiting role during winter (max. 17%, Fig. 6).

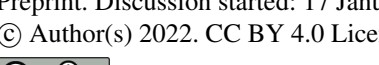



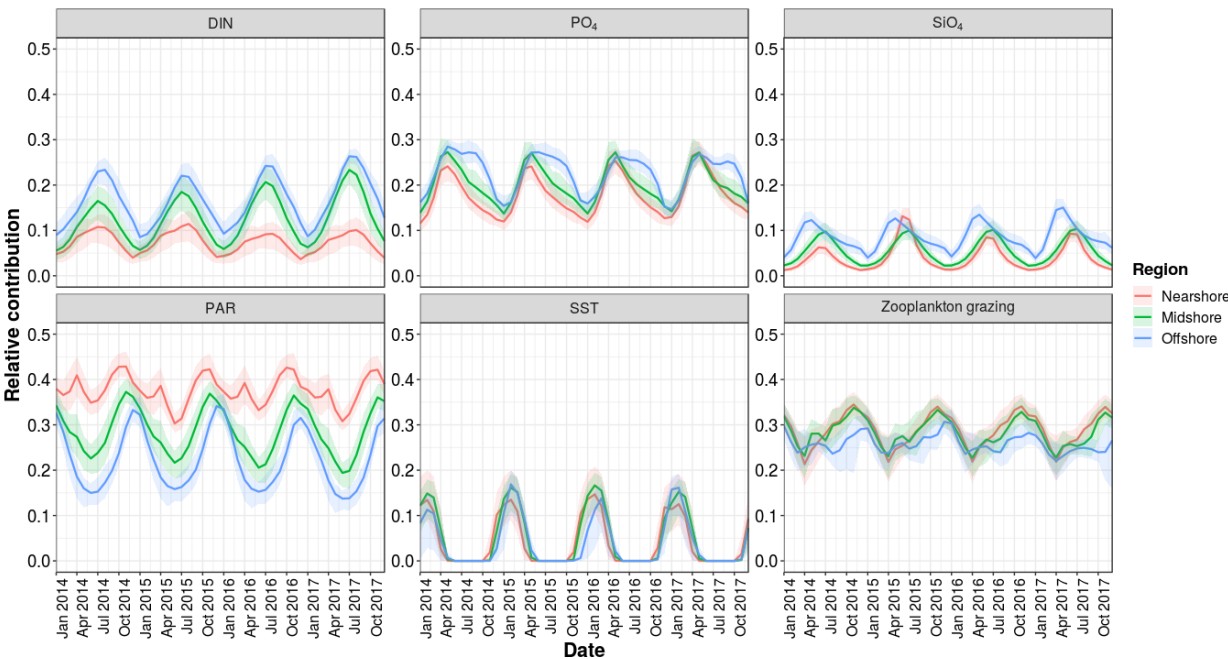

**Figure 6. Monthly averaged relative contributions for each determinant of phytoplankton biomass dynamics for the three regions, i.e. nearshore, midshore and offshore. The shaded areas indicate the 95% confidence interval. Potential determinants are dissolved inorganic nitrogen (DIN), phosphate (PO₄), silicate (SiO₄), solar irradiance (PAR), sea surface temperature (SST), and zooplankton grazing.**

The relative contributions gradually changed along the nearshore–offshore transect (Fig. 6 and 7). For example, PAR's relative contribution is the highest in the nearshore region ($30\% - 43\%$) and decreases towards the open sea ($14\% - 34\%$). Zooplankton grazing is less important in the offshore ($22\% - 31\%$) than in the nearshore region ($21\% - 35\%$) or the midshore region ($23\% - 34\%$). There is also a nearshore–offshore gradient in terms of the relative contribution of nutrients (Fig. 7). For each of the individual nutrients, i.e. DIN, PO₄, SiO₄, we found them more limiting in the offshore region ($8.5\% - 26\%$; $15\% - 29\%$; $3.9\% - 15\%$, respectively) than in the nearshore region ($3.7\% - 11\%$; $12\% - 27\%$; $1.2\% - 13\%$ respectively). SST is equally limiting in all three regions (Fig. 7).





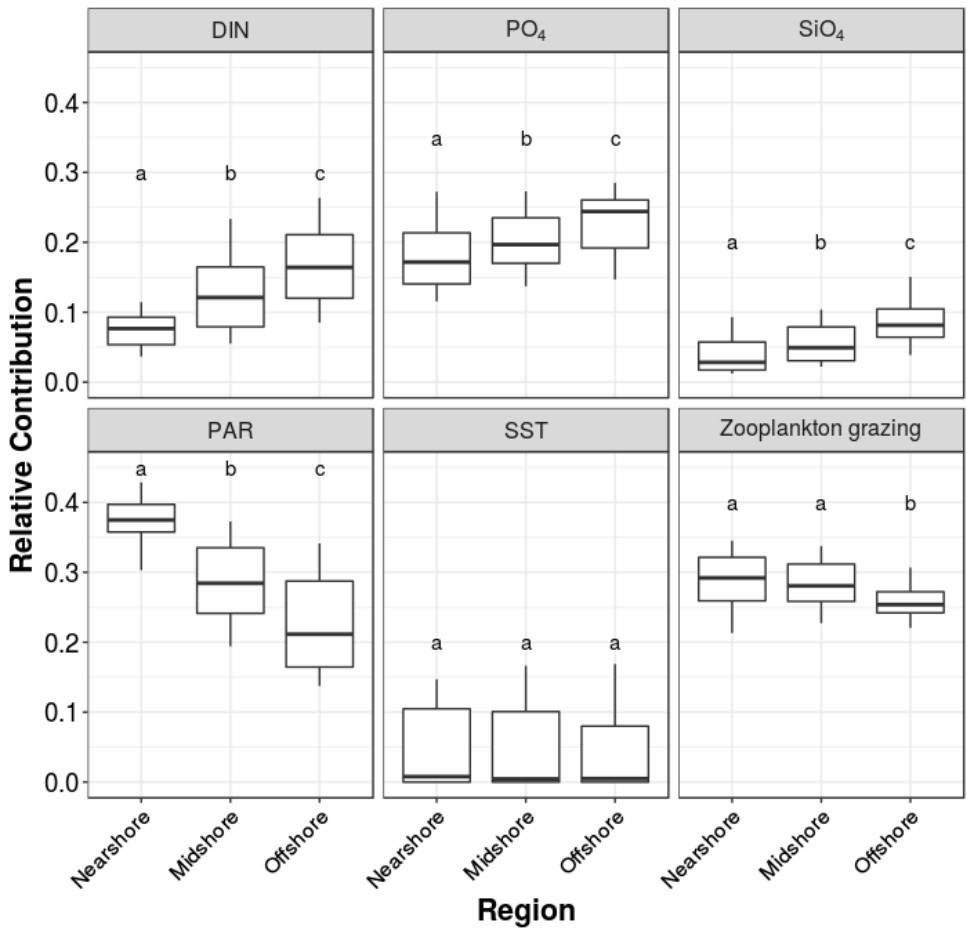

**Figure 7. Monthly averaged relative contributions for each potential determinant of phytoplankton biomass dynamics in nearshore, midshore and offshore region. Statistically significant differences at α = 0.05 between regions are indicated by 'a', 'b', or 'c'. Potential determinants are dissolved inorganic nitrogen (DIN), phosphate (PO$_4$), silicate (SiO$_4$), solar irradiance (PAR), sea surface temperature (SST), and zooplankton grazing.**

## Discussion

Using a NPZ model, we reproduced phytoplankton and zooplankton dynamics in the BPNS (Fig. 4 and 5), and we quantified the relative contribution of the key determinants of phytoplankton biomass, i.e. nutrients, solar irradiance, SST and zooplankton grazing (Fig. 6 and 7). This was done for three regions, i.e. near-, mid- and offshore to examine the spatiotemporal variation. Only a few studies, e.g. Everaert et al. (2015), Llope et al. (2009) and McQuatters-Gollop et al. (2007), have quantified the temporal relative contribution of phytoplankton biomass' key determinants. We found clear regional differences and seasonal patterns in the relative contribution of the key determinants of phytoplankton biomass dynamics.





## 4.1 Comparison phyto- and zooplankton modelling results

Our observed chlorophyll-a concentrations (nearshore: 3.5 – 23.1 mg Chla m$^{-3}$, midshore: 1.8 – 8.8 mg Chla m$^{-3}$, and offshore: 1.4 – 5.3 mg Chla m$^{-3}$), as well as the NPZ model predictions based on these data, are in line with findings of both modelling, e.g. Arndt et al. (2011) and Lancelot et al. (2005), and field studies, e.g. Desmit et al. (2020) and Muylaert et al. (2006), in the BPNS (Fig. 8). In neighbouring regions, similar chlorophyll-a values were observed by e.g. Alvarez-Fernandez and Riegman (2014), Lancelot et al. (2005), European Environment Agency (2019), Colella et al., 2016 and Lundsør et al. (2020; Fig. 8).

Although the chlorophyll-a concentrations are in line with previous studies, we noticed that there is some variability between previous studies and our observations. A potential reason for this is that the data used in this study is more recent. Indeed, with the inclusion of recent observations, Desmit et al. (2020) found a decrease in the annual mean chlorophyll concentration for offshore regions over a time span of 40 years. This is supported by Xu et al. (2020), demonstrating a decreasing trend in chlorophyll-a in the offshore region of the central North Sea.

In the BPNS, our model indicated a clear seasonal pattern with low phytoplankton biomass in winter (min. 1.4 mg Chla m$^{-3}$), and increasing phytoplankton biomass during spring. This spring bloom (max. 23.1 mg Chla m$^{-3}$), typically consisting of diatoms and *Phaeocystis* spp. (Muylaert et al., 2006), occurs in March and April and is followed by a smaller bloom in autumn (max. 7.7 mg Chla m$^{-3}$). We found a decrease in phytoplankton biomass overall, and a decrease of the amplitude of the spring bloom (23.1 mg Chla m$^{-3}$ nearshore to 5.3 mg Chla m$^{-3}$ offshore) with increasing distance to the coast (Fig. 4). These regional

differences were also observed by Desmit et al. (2020; 20 mg Chla m$^{-3}$ nearshore to 12 mg Chla m$^{-3}$ offshore) and Muylaert et al. (2006; 60 mg Chla m$^{-3}$ nearshore to 15 mg Chla m$^{-3}$ offshore) in the BPNS. Muylaert et al. (2006) and Desmit et al. (2020) have also observed that the seasonal pattern, i.e. a spring bloom followed by a smaller autumn bloom, was more distinct closer to the coast (Fig. 4). Jiang et al. (2020) found a high interannual variability in the peak biomass, which is also observed in our field observations, but is less expressed in our modelling results. Nevertheless, in nearshore regions we see that the autumn

bloom is more modest, i.e. three to four times smaller in amplitude, than the spring bloom.

  The classic bimodal bloom pattern that we (Fig. 5) and others (e.g. Lancelot et al. (2005) and Muylaert et al. (2006)) have observed in the BPNS was not found by Nohe et al. (2020) between 2003 and 2010. They found that the spring bloom was more intense and extended, and that there was no autumn bloom. Nohe et al. (2020) suggest that increased SST and water

transparency, and changes in nutrient concentrations and ratios are potential reasons for the lack of an autumn bloom. This lack of an autumn bloom is in strong contrast with our findings with more recent data, i.e. 2014 – 2017. Similar to our study, Speeckaert et al. (2018) did find two blooms, i.e. a spring bloom followed by a smaller autumn bloom in 2016. Whereas Nohe et al. (2020) observed a mean diatom cell density of 3.9 • 10$^5$ cell l$^{-1}$ in autumn in the period 2003 – 2010, Speeckaert et al. (2018) found a peak cell density of 2.0 • 10$^6$ cell l$^{-1}$ with *Guinardia* spp. being the dominant diatom in the autumn bloom in

2016. This autumn bloom is also observed in data from the LifeWatch Flowcam (VLIZ, 2021b) for 2017 and later (Fig. H1, H2 and H3). This suggests that the BPNS may have shifted back to a two-bloom pattern.





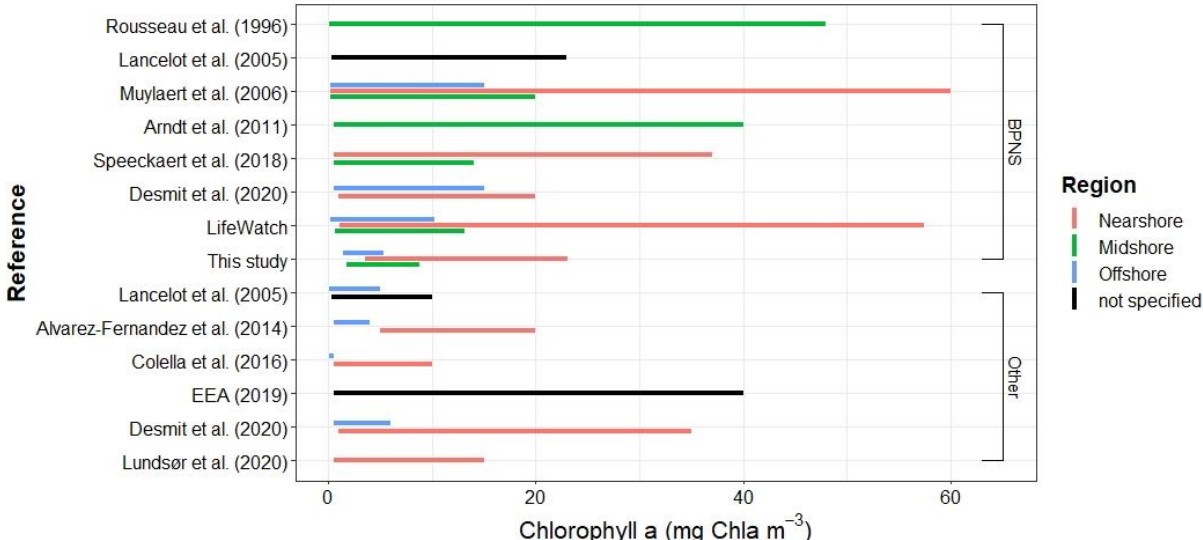

**Figure 8. An overview of the phytoplankton biomass in the Belgian part of the North Sea and other regions, e.g. Netherlands, France, Skagerrak and Mediterranean Sea, aggregated from literature and this study, i.e. simulated phytoplankton biomass and LifeWatch observations.**

The seasonal dynamics of zooplankton correspond largely with data found by Deschutter et al. (2017), Van Ginderdeuren et al. (2014) and Mortelmans et al. (2021) for copepods which largely dominate the zooplankton community in the BPNS (Brylinski, 2009; Van Ginderdeuren et al., 2014). The delayed increase in zooplankton density corresponding with the spring bloom, illustrating the zooplankton grazing on phytoplankton in our model (Fig. 5 and F2; Lancelot et al., 2005), depicts the population dynamics of a classic predator–prey relationship (Wright, 1958). Zooplankton grazing is an important determinant of phytoplankton biomass dynamics, but the predator–prey relationship is complex and changes under different abiotic conditions (Behrenfeld and Boss, 2018).

The LifeWatch observations and our model results suggest higher zooplankton density in nearshore regions (Fig. 4), agreeing well with the findings of Mortelmans et al. (2021) and of Deschutter et al. (2017) for the BPNS. This was also observed in other areas in the world (Leitão et al., 2019; Moore and Sander, 1979). Van Ginderdeuren et al. (2014) observed that the highest copepod densities occurred in their 'midshore region' of the BPNS, but that region overlaps with our nearshore region (Fig. 1).

**4.2 Relative contribution of the key determinants**

The relative contribution of the determinants changes seasonally (Fig. 6). During autumn and winter, when there is the least amount of light in the BPNS (± 8h per day), PAR is the major determinant (max. 43%). Obviously, sufficient solar irradiance is needed for photosynthesis (Reece et al., 2011). SST becomes more limiting for phytoplankton growth during mid-winter and early spring (max. 17%). This was also found by Everaert et al. (2015). They found that the combined relative contribution of SST and PAR varied from 20% during summer to 50% during winter, which is similar to our results (summer: 14% and





winter: 51%). In shallow coastal areas like the BPNS, SST is a key factor affecting phytoplankton bloom dynamics (Trombetta

et al., 2019) as the low temperature decreases the growth rate of marine phytoplankton (Edwards et al., 2016). In early spring when solar irradiance and temperature increases, phosphorus becomes depleted due to the spring bloom (Arndt et al., 2011; Lancelot et al., 2005; van der Zee and Chou, 2005) and becomes the major determining factor (29%). Nitrogen becomes increasingly limiting in early summer (26%), together with silica in the nearshore (13%) and midshore (10%) regions. The increased phytoplankton density during spring bloom results in a higher zooplankton density after the spring bloom in early

May (Mortelmans et al., 2021). Together with zooplankton density, the grazing pressure on phytoplankton increases. During summer and autumn, zooplankton grazing is an important determinant with a high relative contribution on the phytoplankton biomass (31% and 35%, respectively) as zooplankton biomass remains high (Mortelmans et al., 2021) whereas phytoplankton biomass decreases, changing the phytoplankton–zooplankton ratio. This correspond with the findings of Gowen et al. (1999) for the Irish Sea, who found that the percentage of phytoplankton biomass grazing is highest after spring bloom in May. From

the winter onwards, zooplankton grazing's relative contribution decreases as their abundance has decreased together with their food supply, i.e. phytoplankton, which is mainly limited again by the low solar irradiance completing the seasonal cycle of the determinants (Fig. 6).

Besides their temporal variability (cfr. previous paragraph), the key determinants affecting phytoplankton biomass also vary

spatially. The influence of nutrients on the phytoplankton biomass dynamics increases with distance to the coastline (Fig. 7). The main reason for this is related to the fact that nearshore regions tend to be nutrient rich due to riverine discharges. As the river runoff creates a nearshore–offshore gradient in nutrients with lower nutrient availabilities in offshore regions (Fig. E1, Arndt et al., 2011; Van Der Zee and Chou, 2005), all nutrients have a higher relative contribution in limiting phytoplankton biomass (Fig. 6 and 7), i.e. nitrogen (8.5% – 26%), phosphorus (15% – 29%) and silica (3.9% – 15%). The difference in

relative contributions between nearshore and offshore is most obvious for nitrogen, i.e. DIN being more limiting offshore (max. 26%) than nearshore (max. 11%). Phosphorus on the other hand is limiting phytoplankton biomass in offshore regions for a longer period. This could be related to the water depth as phosphorus is largely regenerated from the sediment and offshore there is a much higher water mass to sediment ratio (van der Zee and Chou, 2005). Silica becomes limiting earlier in the offshore region than in the near- and midshore region, as the low $SiO_4$ reserve in the offshore region is depleted quickly

during the spring bloom (Muylaert et al., 2006). The gross part of riverine nutrient resources (Scheldt, Rhine, Meuse and Seine; Lacroix et al 2007) is depleted before it reaches the offshore region (Arndt et al., 2011). Our results largely agree with the findings of Burson et al. (2016), i.e. nearshore regions in the North Sea are P limited and, mid- and offshore regions of the BPNS are N and P co-limited. The impact of zooplankton grazing on phytoplankton biomass is smaller in offshore regions, which could be due to the lower zooplankton density (Fig. F1) and thus lower grazing pressure. In the nearshore region, PAR

is a major determinant year-round (30% – 43%), whereas in mid- and offshore regions (14% – 34%) this is more in balance with the other determinants (Fig. 7). The high turbidity in the nearshore region (Fettweis and Van Den Eynde, 2003; Lacroix





et al., 2007) is restricting phytoplankton's access to sunlight and causes PAR to be a major determining factor throughout the year. The more offshore, the less influential PAR is in terms of phytoplankton biomass dynamics.

Overall, the relative contributions of the key determinants of phytoplankton biomass dynamics showed a clear spatiotemporal variation. Quantifying the relative contribution and identifying the spatiotemporal variation offer a better understanding of how key determinants' limitations to phytoplankton biomass will change under changing conditions, e.g. related to climate change. However, every method has its advantages and limitations, and below we consider those of our modelling approach before addressing potential implication of this study.

### 360 4.3 Modelling with field data: advantages and limitations

Often multiple driver-related research has been performed in laboratories. A big advantage of this is that model species can be kept in optimal conditions, e.g. temperature, light, nutrients, etc., to isolate the effects of the stressor in question. However, these optimal conditions are rarely experienced by organisms in their natural environment (Holmström et al., 2005), hampering the conversion of the laboratory-based conclusions towards field conditions. Therefore, we have selected a modelling approach 365 using field data, which has the advantage that the natural background variation of bottom-up drivers of marine ecosystems are implicitly included in the results (Coull and Chandler, 1992), So, by using field data to quantify the relative contribution, we can assess the impact of the determinants under their natural and continuously changing conditions.

The phytoplankton biomass dynamics modelled in this study agree well with the field observations from LifeWatch (Fig. 4). 370 We acknowledge that the periods with low phytoplankton growth in the offshore region correspond less well with the field observations. This is likely attributable to the seasonal sampling strategy in that region, i.e. no monthly measurements (Mortelmans et al., 2019). As such, we had less data available for the offshore region to calibrate our model.

For the NPZ model, different functional groups of phytoplankton were grouped and by doing so, we may have missed species–specific limitations. It is known that reactions to changes in determinants, such as with climatic changes, are quite species 375 specific (Berdalet et al., 2007; Schlüter et al., 2012). Also, the variation in C:Chla ratio under different environmental conditions was not taken into account (Jakobsen and Markager, 2016). This may partially explain the inability to model the lower chlorophyll-a concentration observed during the winter period as C:Chla ratio describes a seasonal pattern in temperate coastal waters and is related to nutrients (Jakobsen and Markager, 2016). Looking at the nearshore–offshore gradient in this study, another gradient, i.e. east–west gradient, may have been overlooked. The latter may alter the relative contribution of 380 phytoplankton biomass' determinants regarding the distance from the Scheldt estuary. Other studies found a southeast–northwest gradient in nutrients and phytoplankton biomass (Lancelot et al., 2005; Muylaert et al., 2006).





## 4.4 Future perspectives and implications

Our model is well suited to simulate the phyto- and zooplankton dynamics and could be used to fill in data gaps in the measuring of phytoplankton and zooplankton. Besides, our model and results could be used to evaluate how the key limitations
of phytoplankton biomass change under climate change conditions.

During the Anthropocene, human activities have caused ocean warming (IPCC, 2019), acidification (IPCC, 2019) and eutrophication (Paerl et al., 2006) resulting in an on average increasing SST, decreasing pH levels and increased nutrient concentrations compared to the pre-industrial times. To date, it is not clear which long-term changes in abiotic conditions are
most important in the context of marine phytoplankton biomass dynamics. It is hypothesized that the relative contribution of the dominant determinants will change due to changing environmental conditions. According to the IPCC forecasts, the marine environment is expected to warm, and consequently become more acidic (IPCC, 2021). Further, they predict that SST will be less limiting for phytoplankton biomass in our area, while nutrients will become more limiting (IPCC, 2019). Regarding solar irradiance limiting phytoplankton biomass, no change is expected (IPCC, 2019). Although, it can be expected that the
magnitude of the relative contribution of a determinant may in- or decrease under influence of another phytoplankton biomass determining factor. For example, zooplankton grazing on phytoplankton is expected to increase with the warming of seawater (Everaert et al., 2018; IPCC, 2019) and as such, the relative contribution of zooplankton grazing is expected to increase. These changes in environmental abiotic conditions may have a significant impact on the marine phytoplankton biomass (Capuzzo et al., 2018). Indeed, based on field data (Desmit et al., 2020), modelling results (Xu et al., 2020) and laboratory experiments
(Edwards et al., 2016), it is found that changing abiotic environmental conditions impact the marine phytoplankton diversity and the functioning of marine phytoplankton. Modelling and quantifying the relative contribution of phytoplankton's determinants provides a more holistic view rather than a one-to-one relation that is obtained from directly measuring phytoplankton biomass. The quantification of relative contribution, could give more insight in the underlying mechanisms, e.g. changes in the relative contribution of the key determinants of phytoplankton biomass may indicate a disruption of the
phytoplankton community, such as a change in community structure (Benedetti et al., 2021; Ferreira et al., 2020).

Even though phytoplankton biomass dynamics are resilient to a certain extent (Wiltshire et al., 2008), Wells et al. (2020) stressed the importance of having a better understanding of the effects of each of the key determinants on various subgroups of phytoplankton in order to predict whether the ecosystem will change. We provided a first step in this direction with our
NPZ model that has the potential to be further developed to a more detailed taxonomical level.



## Conclusion

In this study, we quantified phytoplankton biomass dynamics and the relative contribution of its determinants using a NPZ model along a near–offshore gradient and a period of four years (2014 – 2017). By doing so, a better understanding of the spatiotemporal variations in these contributions is provided. We found that the relative contributions of phytoplankton
determinants alter spatially and temporally. Solar irradiance (up to 43%) and zooplankton grazing (up to 35%) are the most influential determinants for phytoplankton biomass and this throughout the year. A clear spatial gradient was observed for most of the determinants, e.g. nutrients and zooplankton grazing are more limiting offshore, while the opposite is true for solar irradiance. We suggest modelling and quantifying the relative contribution of phytoplankton determinants to have a better understanding of the effects of each of the key determinants on phytoplankton in order to predict whether the ecosystem will
change under future climate scenarios and/or blue economy activities.

## Appendix A. Nutrient–Phytoplankton–Zooplankton Model Equations

We simulated phytoplankton and zooplankton abundances from 2014 to 2017 using the nutrient–phytoplankton–zooplankton–detritus (NPZD) ecosystem model proposed by Soetaert and Herman (2009) and adjusted by Everaert et al. (2015), i.e. a nutrient–phytoplankton–zooplankton (NPZ) model.

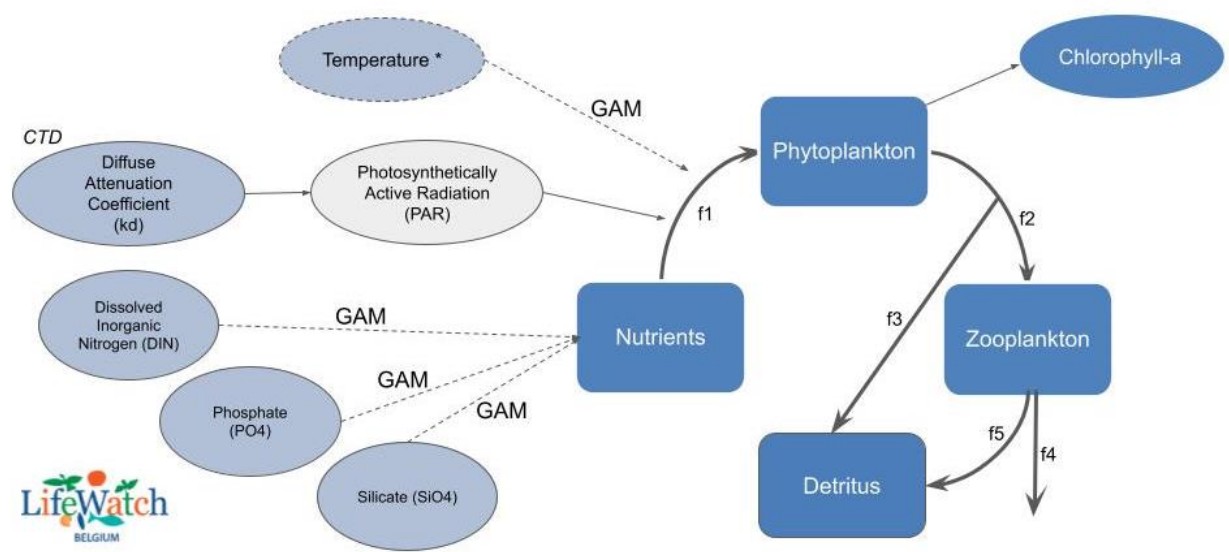


**Figure A1. Nutrient–phytoplankton–zooplankton–detritus (NPZD) model structure. Based on Soetaert and Herman (2009) and Everaert et al. (2015). *Sea surface temperature data were generated using generalized additive models in nearshore and midshore stations. Offshore stations SST data were retrieved from the Westhinder station from the Flemish Banks Monitoring Network.**

The equations that determine the rates of change in the abundance of nutrients, phytoplankton (PHYTO) and zooplankton
(ZOO) in the model are:



$$dPHYTO/dt = Nuptake\ (f1) - Grazing\ (f2)$$

$$dZOO/dt = Grazing\ (f2) - Faeces\ (F3) - Excretion\ (f4) - Mortality\ (f5)$$

where t indicates time in days, PHYTO phytoplankton, ZOO zooplankton. Daily abundance of nutrients is determined as the
total sum of DIN, $PO_4$ and $SiO_4$ based on the generalized additive models (GAM) of each type of nutrient.

$$NUTRIENTS = GAM\ (DIN) + GAM\ (PO4) + GAM\ (SiO_4)$$

$$Nuptake\ (f1) = maxUptake * PAR\_lim * Temp\_lim * P\_lim * DIN\_lim * Si\_lim *PHYTO$$
$$Grazing\ (f2) = maxGrazing* (PHYTO/(PHYTO+ksGrazing))*ZOO$$

$$Faeces\ (f3) = pFaeces * Grazing$$

$$Excretion\ (f4) = excretionRate * ZOO$$

$$Mortality\ (f5) = mortalityRate * ZOO^2$$

$$Chlorophyll = chlNratio * PHYTO$$


where Nuptake is the phytoplankton nitrogen uptake, PAR_lim the limitation factor for PAR, Temp_lim for sea surface
temperature (SST), P_lim for $PO_4$, DIN_lim for DIN and Si_lim for $SiO_4$. Parameters of the model are maxUptake,
mineralizationRate, excretionRate, maxGrazing, klGrazing, pFaeces, mortalityRate and ChlNratio.

The equations used to define the limitation factors follow saturating Michaelis–Menten equations (Soetaert and Herman, 2009).
These equations are:

$$PAR\_lim = PAR/(PAR+ksPAR)$$

$$Temp\_lim = theta\char`^(Temp - Tobs)$$

$$DIN\_lim = DIN /(DIN+ksDIN)$$

$$P\_lim = P/(P + ksP)$$

$$Si\_lim = Si/(Si + ksSi)$$

$$PAR\ (f7) = I_0\ e^{\ -kd * z}$$

$$Theta = 1.185 - 0.00729*Temp$$

where ksPAR, Tobs, ksDIN, ksP and ksSi are parameters of the model.

The equation for PAR (f7) is defined based on the Lambert-Beer law (Kirk, 1994; Lund-Hansen, 2004):

$$PAR\ (f7) = I_0\ e^{\ -kd * z}$$



Where $I_0$ is the surface irradiance ($\mu$Einst m$^{-2}$ s$^{-1}$), z depth (m), and $k_d$ the diffuse attenuation coefficient (m$^{-1}$).

In the model, the surface irradiance is modelled as photosynthetically active radiation defined in Soetaert and Herman (2009):

$$I_0 = 0.5*(540+440*\sin(2*pi*t/365-1.4))$$

where t indicates the day of the year.

The diffuse attenuation coefficient ($k_d$) describes the rate at which light diminishes with depth due to absorption and scattering in the water column (Devlin et al., 2009; Kirk, 1994; Lund-Hansen, 2004). $K_d$ is used as a proxy of the influence of SPM on PAR (Devlin et al., 2009). $K_d$ values were calculated based on the PAR and depth data recorded in the Marine Information and

Data Acquisition System (MIDAS).

Depth (z) used in equation f7 is 3m as LifeWatch data is measured at 3m depth (Mortelmans et al., 2019).

The NPZ model and corresponding calculations were performed in R (R Core Team, 2018; version 3.4.4) using the following

packages 'doParallel' (Microsoft Corporation and Weston, 2020), 'dplyr' (Wickham et al., 2020), 'foreach' (Microsoft and Weston, 2020), 'ggplot2' (Wickham, 2016), 'ggpubr' (Kassambara, 2019), 'lubridate' (Grolemund and Wickham, 2011), 'parallel' (R Core Team, 2018), 'plyr' (Wickham, 2011), 'RColorBrewer' (Neuwirth, 2014), 'reshape2' (Wickham, 2007), 'stats' (R Core Team, 2018), 'viridis' (Garnier, 2018), 'xts' (Ryan and Ulrich, 2020).

### Appendix B. NPZD Model Parameters

**Table B1. Possible value parameters for the nutrient–phytoplankton–zooplankton (NPZ) model based on literature. These intervals are used as a reference to create the different sets to calibrate the NPZ model.*Minimum and maximum values for $k_d$ were calculated for each region of interest based on CTD data recorded in MIDAS.**

| Parameter | Minimum possible value | Maximum possible value | References |
|---|---|---|---|
| maxUptake | 0.25 (day$^{-1}$) | 1.5 (day$^{-1}$) | Everaert et al. (2015) |
| excretionRate | 0.1 (day$^{-1}$) | 0.2 (day$^{-1}$) | Everaert et al. (2015) |
| maxGrazing | 0.8 (day$^{-1}$) | 1 (day$^{-1}$) | Everaert et al. (2015) |
| ksGrazing | 1 (mmol N m$^{-3}$) | 4 (mmol N m$^{-3}$) | Everaert et al. (2015) |
| pFaeces | 0.2 (day$^{-1}$) | 0.5 (day$^{-1}$) | Everaert et al. (2015) |





| mortalityRate | 0.25 ( (mmol N m$^{-3}$)$^{-1}$ day$^{-1}$) | 0.5 ( (mmol N m$^{-3}$)$^{-1}$ day$^{-1}$) | Everaert et al. (2015) |
|---|---|---|---|
| ChlNratio | 1 (mg chla / mmolN ) | 8 (mg chla / mmolN ) | Alvarez-Fernandez and Riegman (2014) |
| ksPAR | 30 (Einst m$^{-2}$ s$^{-1}$) | 250 (Einst m$^{-2}$ s$^{-1}$) | Everaert et al. (2015) |
| Tobs | 7 °C | 15 °C | Everaert et al. (2015) |
| ksDIN | 0.25 (mmol N m$^{-3}$) | 5 (mmol N m$^{-3}$) | Everaert et al. (2015) |
| ksP | 0.2 (mmol P m$^{-3}$) | 0.5 (mmol P m$^{-3}$) | Everaert et al. (2015) |
| ksSi | 0.2 (mmol Si m$^{-3}$) | 0.8 (mmol Si m$^{-3}$) | Lancelot et al. (2005) |
| Kd* | 0.6 (m$^{-1}$)<br>0.27<br>0.21 | 1 (m$^{-1}$)<br>0.67<br>0.44 | Nearshore station<br>Midshore station<br>Offshore stations |

**Table B2. Root Mean Square Error (RMSE) for each of the regions of interest based on the second iteration, the 10% best simulations**
**which correspond to the number of simulations indicated in the last column. Each region of interest has a different number of best simulations as some combination of set of parameters results in exponential behaviour. Only the simulations that converged were considered to select the 10% best (i.e. lowest RMSE) out of the 5000 possible simulations.**

| Region of interest | RMSE - Median (Q1 – Q3) | Number of simulations |
|---|---|---|
| Nearshore region | 1.34 (1.32 – 1.36) | 259 |
| Midshore region | 0.44 (0.43 – 0.45) | 498 |
| Offshore region | 0.40 (0.37 – 0.41) | 499 |

**Table B3. Number of observations available of Chlorophyll-a and zooplankton densities for the nearshore, midshore and offshore**
**region (Fig. 2) from 2014 to 2017.**

| Variable | Nearshore region | Midshore region | Offshore region |
|---|---|---|---|
| Chlorophyll-a | 39 | 37 | 98 |
| Zooplankton | 40 | 37 | 96 |



## Appendix C. Zooplankton conversion

The most common zooplankton found in the Belgian part of the North Sea (BPNS) are copepods (Van Ginderdeuren et al., 2014). The most common species are *Acartia clausi, Temora longicornis, Paracalanus parvus, Centropages hamatus, Pseudocalanus elongatus, Centropages typicus, Calanus helgolandicus* and *Euterpina acutifrons* (Van Ginderdeuren et al., 2014). The dinoflagellate *Noctiluca scintillans* were also seasonally found in high densities and the Appendicularia *Oikopleura dioica* was found year round (Van Ginderdeuren et al., 2014). Therefore, the taxa Calanoida, Noctiluca, Harpacticoida and Appendicularia were selected from the LifeWatch database to calculate the zooplankton abundance (ind m$^{-3}$).

The body mass per individual of each taxon was defined based on the most common species of each group (Table C1). The body mass per individual was calculated as the median value of body mass (mg C ind$^{-1}$) for the most common species of each taxon. Afterwards, the body mass in Carbon was converted to mmol C by dividing by the molecular weight of C (12.0107 gr / mole). Finally, the mmol C m$^{-3}$ is converted to mmol N m$^{-3}$ based on the C:N ratio of each taxon (Table C2).

**Table C1. Body mass per individual (mg C ind$^{-1}$) of the most common species of taxa found in the Belgian part of the North Sea.**

| Taxon | Body mass (mg C ind$^{-1}$) | Species | Reference |
|---|---|---|---|
| Calanoida | 0.0006 | *Acartia clausi, Temora longicornis, Paracalanus parvus, Centropages hamatus, Pseudocalanus elongatus, Centropages typicus and Calanus helgolandicus* | Brun et al. (2016) |
| Noctiluca | 0.0003 | *Noctiluca scintillans* | Löder et al. (2012) |
| Harpacticoida | 0.001 | *Euterpina acutifrons* | Sautour and Castel (1995) |
| Appendicularia | 0.002 to 0.006 | *Oikopleura dioica* | Lombard et al. (2009) |


**Table C2. C:N ratio for each of the most common taxa present in the Belgian part of the North Sea.**

| Taxon | C:N ratio | Species | Reference |
|---|---|---|---|
| Calanoida | 5.5 – 7 | *Acartia* spp., *Temora* sp., *Centropages, Oithona* sp., *Pseudo/Paracalanus* spp. | Van Nieuwerburgh et al. (2004) |





| Noctiluca | 2.3 – 4.4 | *Noctiluca scintillans* | Tada et al. (2000) |
| Harpacticoida | 4.26 – 4.74 7.7 – 8.1 | *Euterpina acutifrons* | Szyper (1989) Abdel-Moati et al. (1993) |
| Appendicularia | 4.08 | *Oikopleura dioica* | Lombard et al. (2009) |

**Appendix D. Smoothers for generalized additive models (GAM) in the three regions of interest, i.e. near-, mid- and offshore region.**






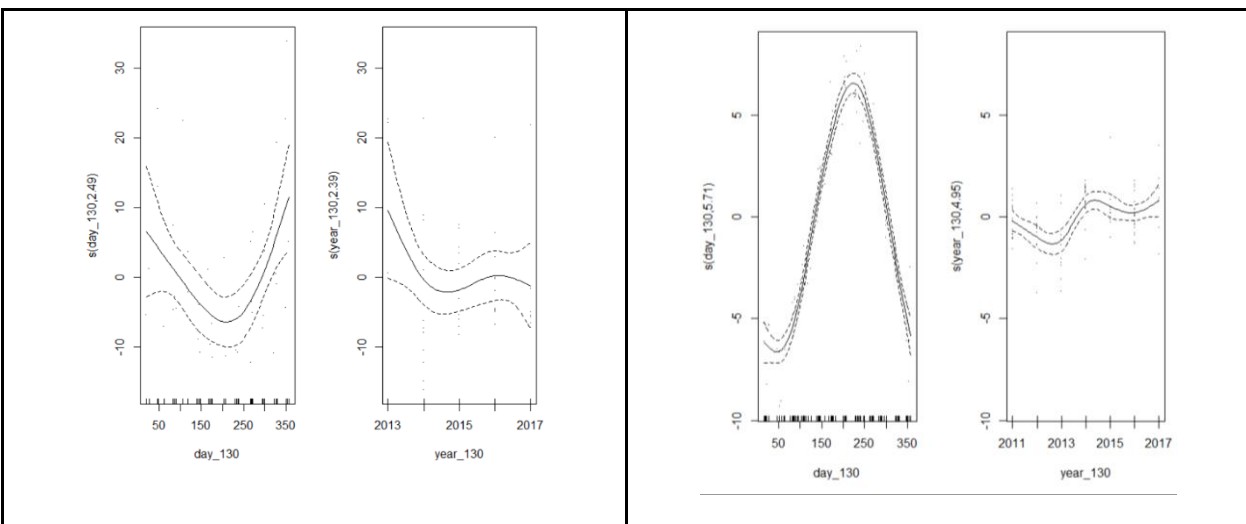

**Figure D1. Smoothers for day and year input variables, i.e. Ammonium (NH₄), Phosphate (PO₄), Nitrite (NO₂), Silicate (SiO₄),**
**Nitrate (NO₃), and sea surface temperature (SST), for generalized additive models in the nearshore region. The smoothers are**
**calculated from 2011 to 2017 as the three first years are used as dummy years.**

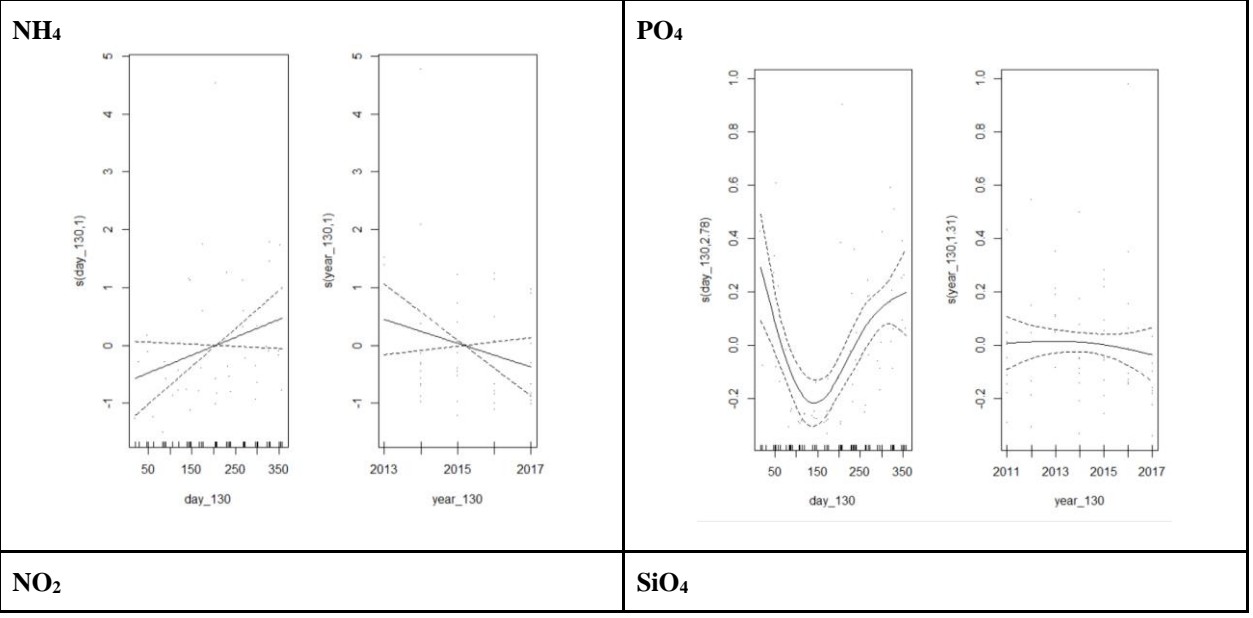





**Figure D2. Smoothers for day and year input variables, i.e. Ammonium (NH₄), Phosphate (PO₄), Nitrite (NO₂), Silicate (SiO₄), Nitrate (NO₃), and sea surface temperature (SST), for generalized additive models in midshore region. The smoothers are calculated from 2011 to 2017 as the three first years are used as dummy years.**

| NH₄ | PO₄ |
|---|---|
|  |  |





**NO₂**

**SiO₄**

**NO₃**





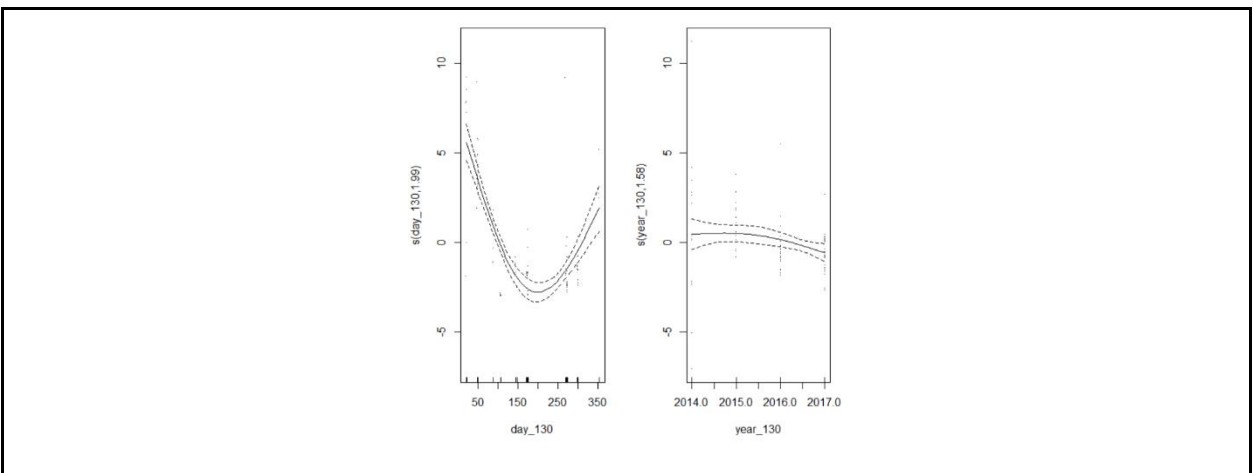

**Figure D3. Smoothers for day and year input variables, i.e. Ammonium (NH₄), Phosphate (PO₄), Nitrite (NO₂), Silicate (SiO₄), and Nitrate (NO₃), for generalized additive models (GAM) in the offshore region. The smoothers are calculated from 2011 to 2017 as the three first years are used as dummy years. Daily sea surface temperature (SST) data were extracted from the Flemish Banks Monitoring Network, no GAM were applied.**

**Appendix E. Time trends GAM in the three regions of interest**

Time trends of nutrient and SST data were created using the GAM models for each region of interest (Appendix D). Detailed comparisons of observations are found in table E1 and E2 and figure E1.

**Table E1. Performance of generalized additive models to create daily time trends of nutrient data, i.e. Dissolved Inorganic Nitrogen (DIN), Phosphate (PO₄) and Silicate (SiO₄), and sea surface temperature (SST) data in the three regions of interest**

| Region of interest | Nutrient And SST | RMSE | $R^2$ |
|---|---|---|---|
| Nearshore region | DIN (mmol N m$^{-3}$) | 9.61 | 0.30 |
| | PO₄ (mmol P m$^{-3}$) | 0.30 | 0.32 |
| | SiO₄ (mmol Si m$^{-3}$) | 6.43 | 0.39 |
| | SST (°C) | 1.82 | 0.94 |
| Midshore region | DIN (mmol N m$^{-3}$) | 5.69 | 0.37 |
| | PO₄ (mmol P m$^{-3}$) | 0.23 | 0.30 |
| | SiO₄ (mmol Si m$^{-3}$) | 3.65 | 0.29 |





| | SST (°C) | 0.88 | 0.96 |
|---|---|---|---|
| Offshore region | DIN (mmol N m⁻³) | 2.33 | 0.55 |
| | PO₄ (mmol P m⁻³) | 0.05 | 0.87 |
| | SiO₄ (mmol Si m⁻³) | 1.30 | 0.30 |

**Table E2. Performance of different generalized additive models for each variable of interest, i.e. Phosphate ($PO_4$), Ammonium ($NH_4$), Nitrite ($NO_2$), Nitrate ($NO_3$), Silicate ($SiO_4$), and sea surface temperature (SST), in nearshore, midshore and offshore using data from 2011 to 2017. Models in bold were selected for modelling the corresponding variable.**

| | | | | | Nearshore region |
|---|---|---|---|---|---|
| **Variable** | **k-s(day)** | **k-s(year)** | **AIC** | **Adjusted R²** | **K performance** |
| PO₄ | 3 | 3 | 58.1 | 0.19 | p-value < 0.05 for day, and 0.38 for year |
| | 4 | 4 | 50.26 | 0.30 | p-value < 0.05 for day, and 0.23 for year |
| | **5** | **5** | **46.96** | **0.35** | p-value < 0.05 for day, and 0.07 for year |
| | 6 | 6 | 47.6 | 0.36 | p-value < 0.05 for day, and 0.06 for year |
| NH₄ | 3 | 3 | 203.54 | 0.04 | p-values > 0.05 for both smoothers |
| | **4** | **4** | **201.27** | **0.12** | p-values > 0.05 for both smoothers |
| | 5 | 5 | 202.06 | 0.11 | p-values > 0.05 for both smoothers |
| NO₂ | 3 | 3 | 44.04 | 0.08 | p-value < 0.05 for day, and 0.86 for year |
| | **4** | **4** | **33.44** | **0.25** | p-value < 0.05 for day, and 0.79 for year |
| | 5 | 5 | 34.31 | 0.25 | p-value < 0.05 for day, and 0.81 for year |
| NO₃ | 3 | 3 | 336.38 | 0.25 | p-values < 0.05 for day, and 0.56 for year |
| | **4** | **4** | **334.55** | **0.31** | p-values < 0.05 for day and 0.26 for year |
| | 5 | 5 | 335.36 | 0.32 | p-values < 0.05 for day, and 0.24 for year. |
| SiO₄ | 3 | 3 | 481.08 | 0.28 | p-value < 0.05 for day, and 0.22 for year |
| | 4 | 4 | 476.2 | 0.35 | p-value < 0.05 for both smoothers |
| | **5** | **5** | **471.5** | **0.41** | p-value < 0.05 for both smoothers |
| | 6 | 6 | 472.28 | 0.41 | p-value < 0.05 for both smoothers |
| | | | | | Midshore region |
| **Variable** | **k-s(day)** | **k-s(year)** | **AIC** | **Adjusted R²** | **K performance** |
| PO₄ | 3 | 3 | 10.82 | 0.21 | p-values < 0.05 for day, and 0.67 for year |
| | **4** | **4** | **4.53** | **0.30** | p-values < 0.05 for day, and 0.71 for year |
| | 5 | 5 | 4.96 | 0.31 | p-values < 0.05 for day, and 0.69 for year |
| NH₄ | 3 | 3 | 136.188 | 0.09 | p-values > 0.05 for both smoothers |
| | **4** | **4** | **136.19** | **0.09** | p-values > 0.05 for both smoothers |
| | 5 | 5 | 136.188 | 0.09 | p-values > 0.05 for both smoothers |





| Variable | k-s(day) | k-s(year) | AIC | Adjusted R² | K performance |
|---|---|---|---|---|---|
| NO₂ | **3** | **3** | **-27.8** | **0.37** | p-values < 0.05 for day, and 0.68 for year |
| | 4 | 4 | -26.36 | 0.37 | p-values < 0.05 for day, and 0.72 for year |
| | 5 | 5 | -25.81 | 0.37 | p-values < 0.05 for day, and 0.72 for year |
| NO₃ | **3** | **3** | **271.38** | **0.37** | p-values < 0.05 for day, and 0.31 for year |
| | 4 | 4 | 273.25 | 0.38 | p-values < 0.05 for day, and 0.41 for year |
| | 5 | 5 | 274.05 | 0.38 | p-values < 0.05 for day, and 0.39 for year |
| SiO₄ | **3** | **3** | **362.68** | **0.28** | p-value < 0.05 for day, and 0.54 for year |
| | 4 | 4 | 363.23 | 0.28 | p-value < 0.05 for day, and 0.56 for year |
| | 5 | 5 | 363.47 | 0.28 | p-value < 0.05 for day, and 0.55 for year |

| Offshore region | | | | | |
|---|---|---|---|---|---|
| **Variable** | **k-s(day)** | **k-s(year)** | **AIC** | **Adjusted R²** | **K performance** |
| PO₄ | 3 | 3 | -281.69 | 0.80 | p-value < 0.05 for both smothers |
| | 4 | 4 | -288.22 | 0.82 | p-values < 0.05 for both smoothers |
| | 5 | 5 | -306.43 | 0.84 | p-values < 0.05 for both smoothers |
| | **6** | **6** | **-327.19** | **0.87** | p-values < 0.05 for both smoothers |
| NH₄ | 3 | 3 | 232.78 | 0.05 | p-values >= 0.05 for both smoothers |
| | **4** | **4** | **230.73** | **0.09** | p-values > 0.05 for both smoothers |
| NO₂ | 3 | 3 | -46.83 | 0.37 | p-values < 0.05 for both smothers |
| | 4 | 4 | -72.92 | 0.51 | p-values < 0.05 for both smothers |
| | 5 | 5 | -92.81 | 0.60 | p-values < 0.05 for both smothers |
| | **6** | **6** | **-125.85** | **0.70** | p-values < 0.05 for both smoothers |
| NO₃ | **3** | **3** | **430.44** | **0.62** | p-values < 0.05 for day, and 0.10 for year |
| | 4 | 4 | 431.57 | 0.62 | p-values < 0.05 for day, and 0.14 for year |
| SiO₄ | 3 | 3 | 401.81 | 0.26 | p-values < 0.05 for day, and 0.07 for year |
| | 4 | 4 | 399.83 | 0.28 | p-value < 0.05 for day, and 0.37 for year |
| | **5** | **5** | **396.58** | **0.31** | p-value < 0.05 for day, and 0.47 for year |
| | 6 | 6 | 396.99 | 0.31 | p-value < 0.05 for day, and 0.47 for year |




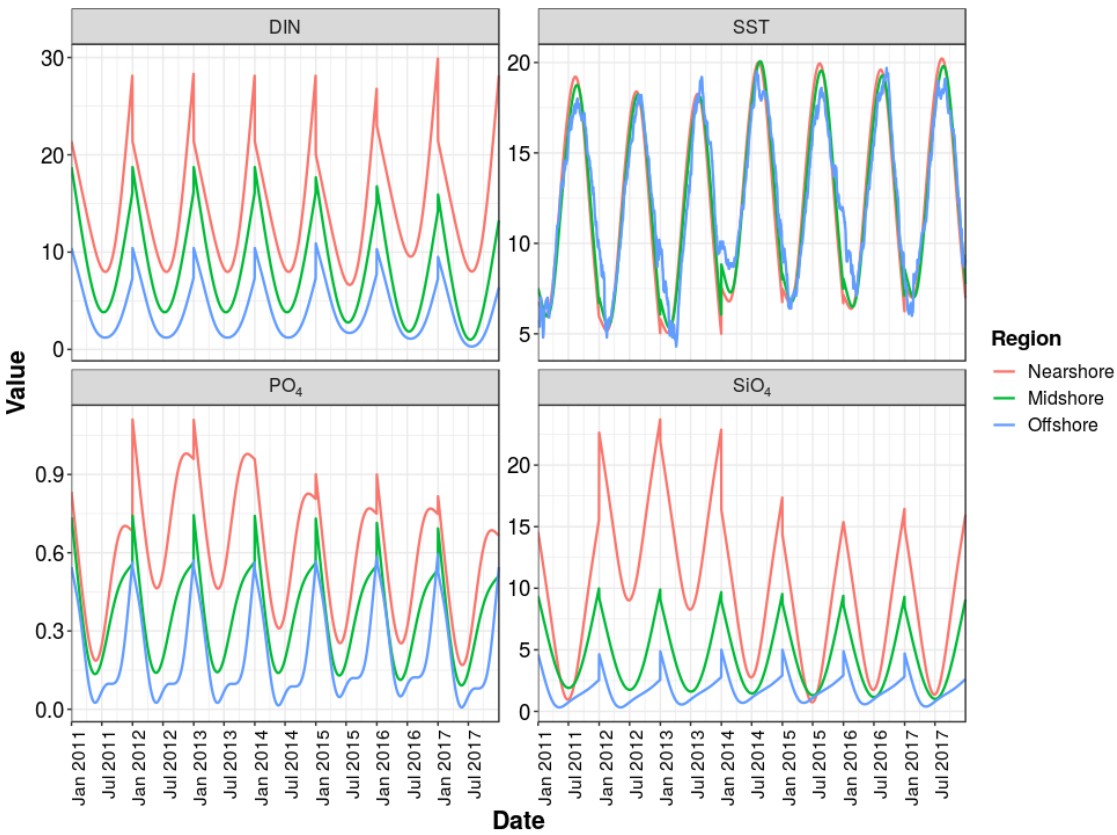

**Figure E1. Time trends created using generalized additive models to generate input data for the nutrient–phytoplankton–zooplankton model. The first three years (2011 to 2013) are used as dummy years to stabilize the initial conditions of the model. These years are removed and final results are only considered from 2014 to 2017.**





## Appendix F. Simulated Zooplankton abundances


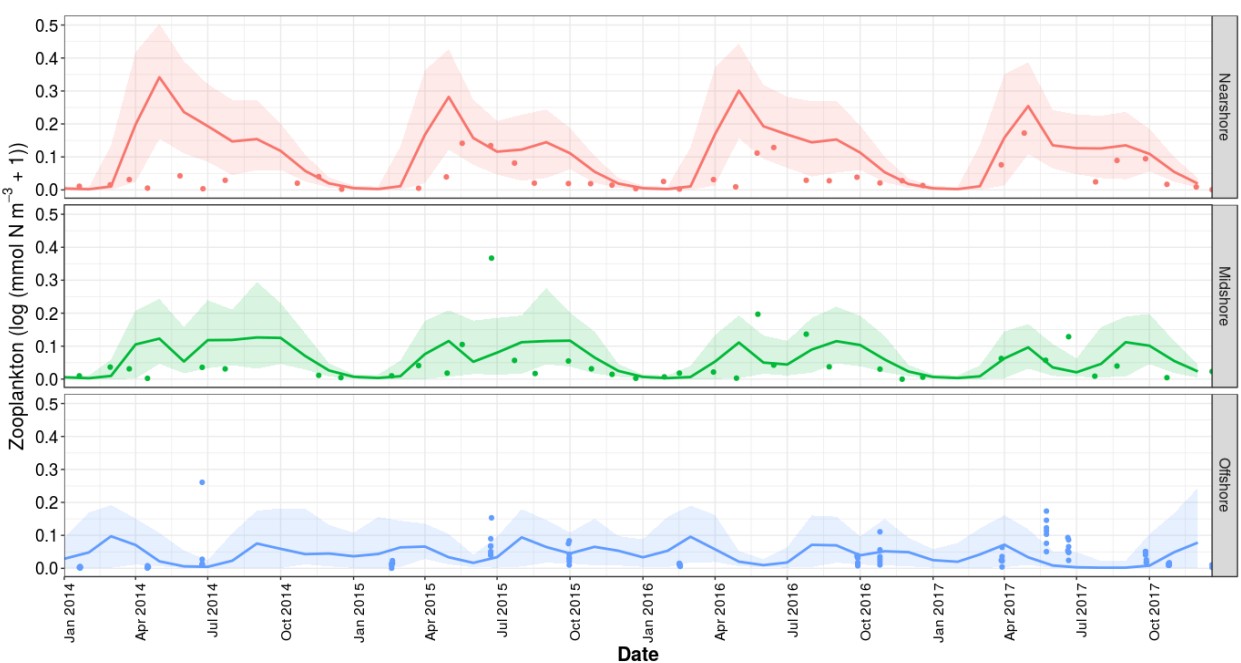

**Figure F1.** Zooplankton density simulations using the nutrient–phytoplankton–zooplankton model in nearshore, midshore and offshore regions in the Belgian part of the North Sea. The bold lines indicate the average zooplankton density predictions and the shaded regions represent the 95% confidence interval. The dots are observed values collected during the LifeWatch campaigns.


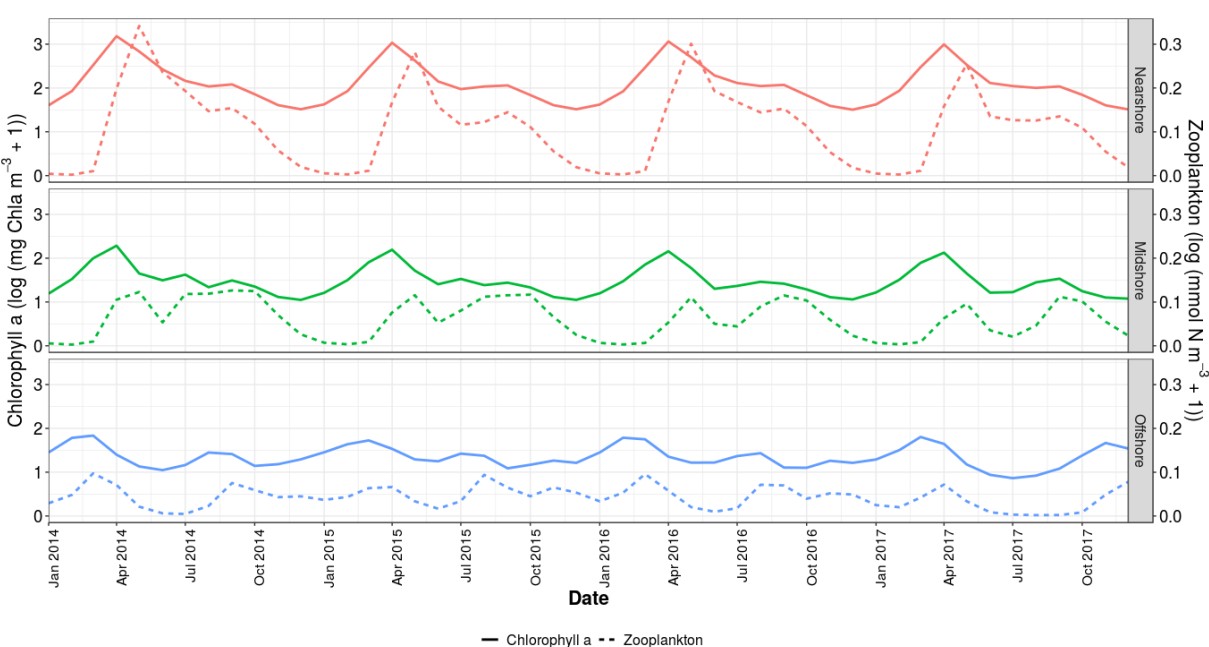





**Figure F2. Phyto- and zooplankton density simulations using the nutrient–phytoplankton–zooplankton model in nearshore, midshore and offshore regions in the Belgian part of the North Sea. The lines represent the average phyto- and zooplankton density predictions.**

**Appendix G. The relative contributions of the determinants of phytoplankton biomass dynamics in the near-, mid- and offshore region**

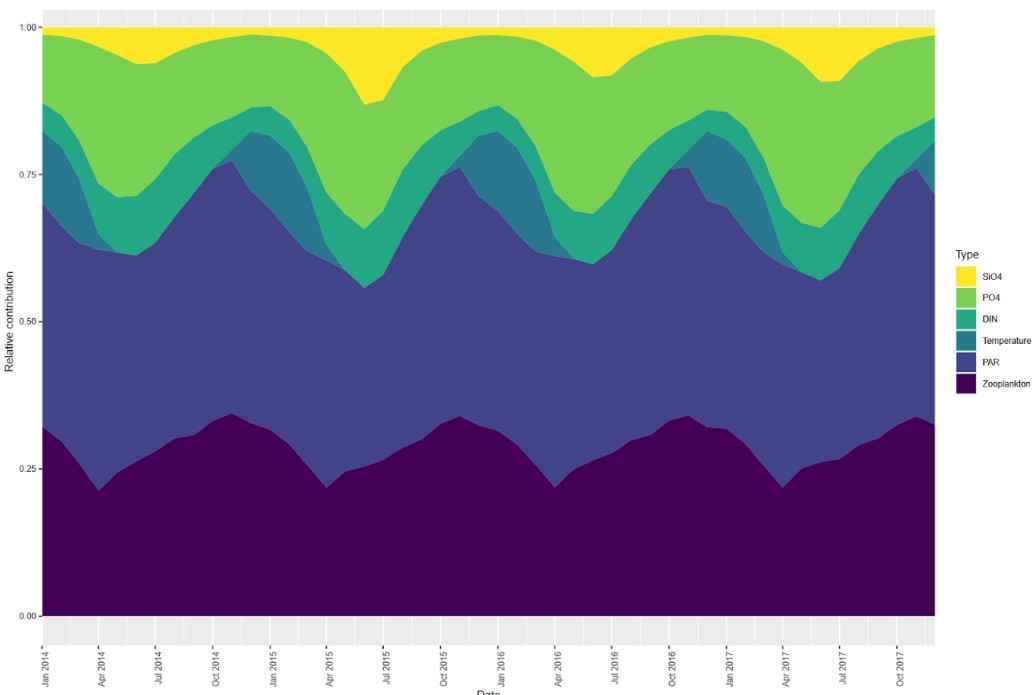

**Figure G1. Average monthly relative contributions for each determinant of phytoplankton biomass dynamics in the nearshore region. Potential determinants are dissolved inorganic nitrogen (DIN), phosphate (PO₄), silicate (SiO₄),** 555 **photosynthetically active radiation (PAR), sea surface temperature (SST), and zooplankton grazing.**





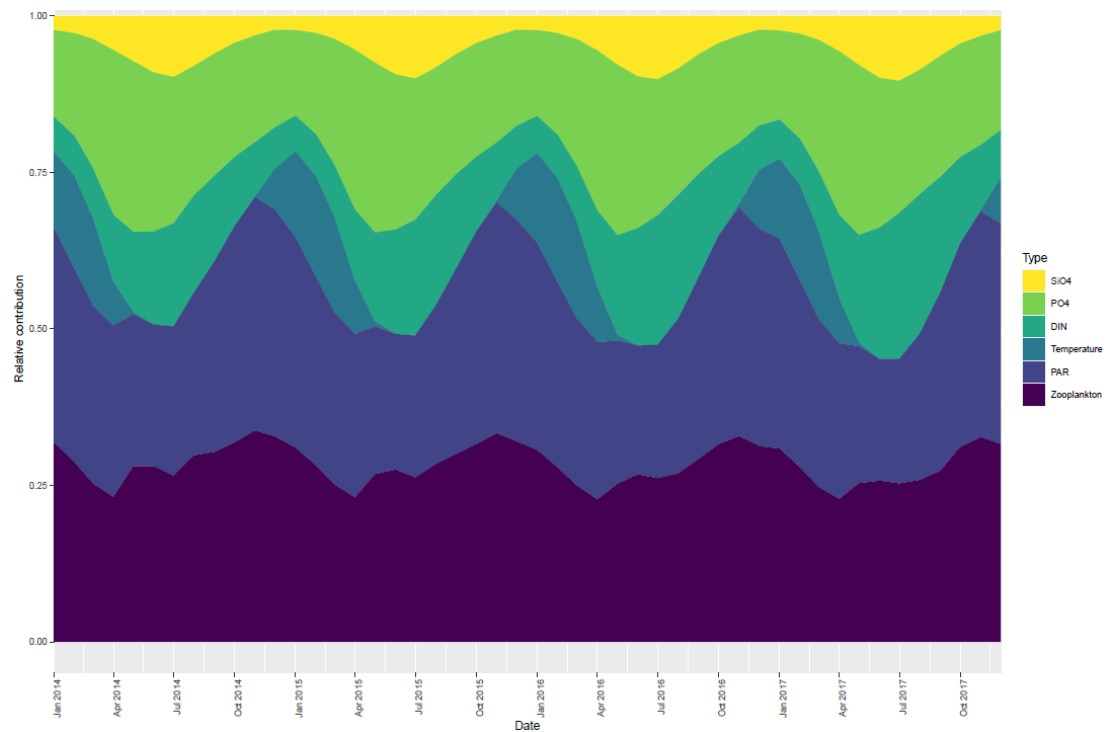

**Figure G2. Average monthly relative contributions for each determinant of phytoplankton biomass dynamics in the midshore regions. Potential determinants are dissolved inorganic nitrogen (DIN), phosphate (PO₄), silicate (SiO₄), photosynthetically active radiation (PAR), sea surface temperature (SST), and zooplankton grazing.**






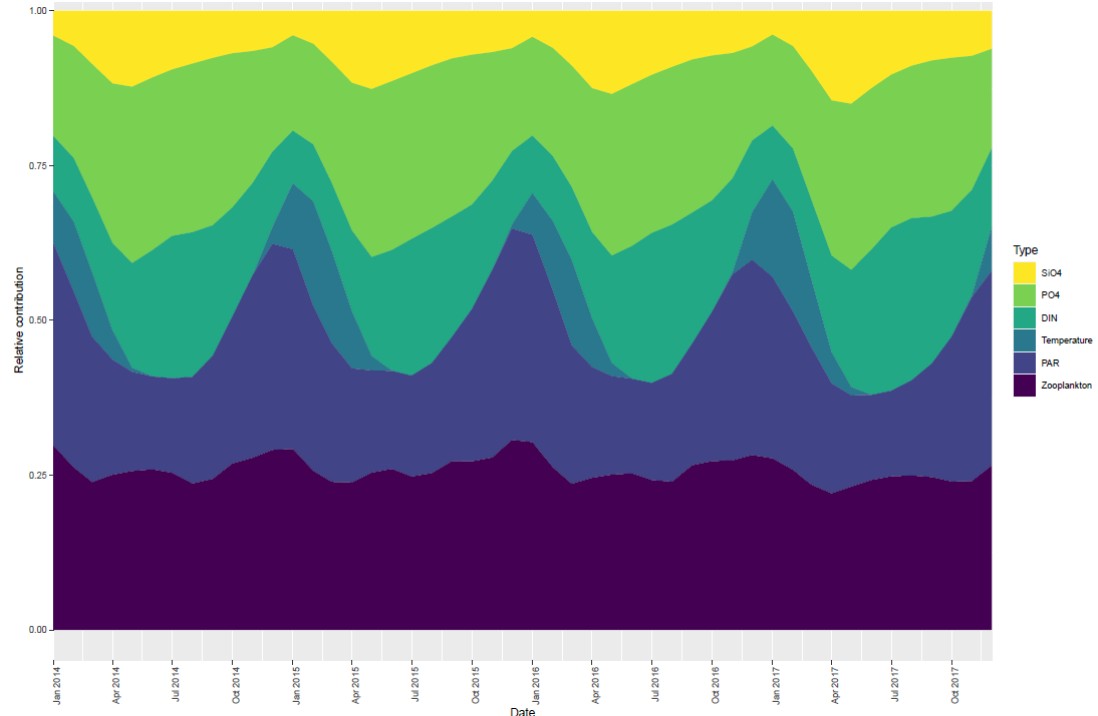

**Figure G3.** Average monthly relative contributions for each determinant of phytoplankton biomass dynamics in the offshore region. Potential determinants are dissolved inorganic nitrogen (DIN), phosphate (PO₄), silicate (SiO₄), photosynthetically active radiation (PAR), sea surface temperature (SST), and zooplankton grazing.




**Appendix H. Diatom cell density in 2017**

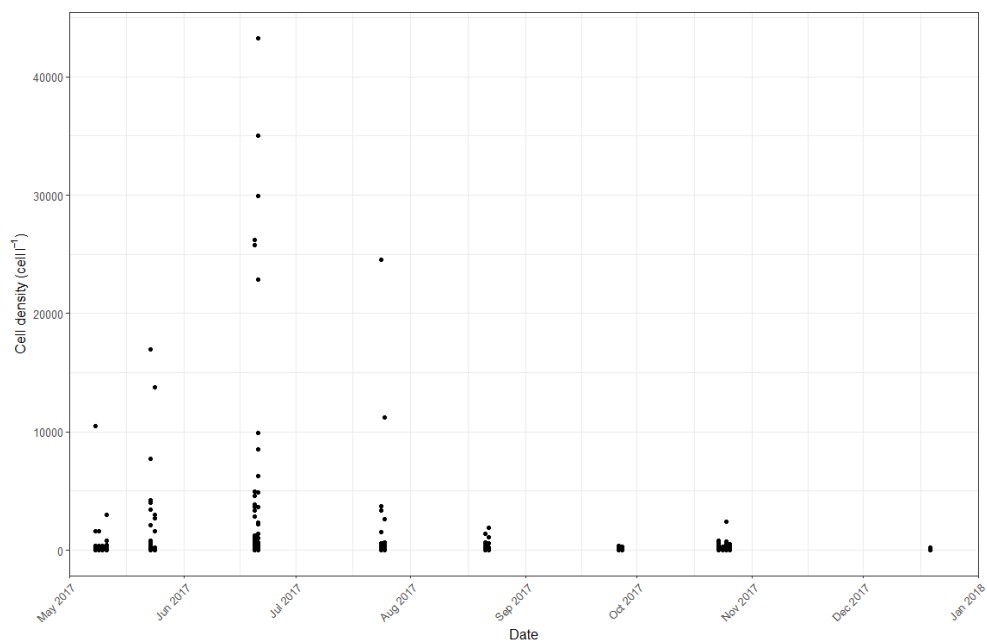

**Figure H1. Diatom cell density in 2017 observed with the LifeWatch Flowcam.**

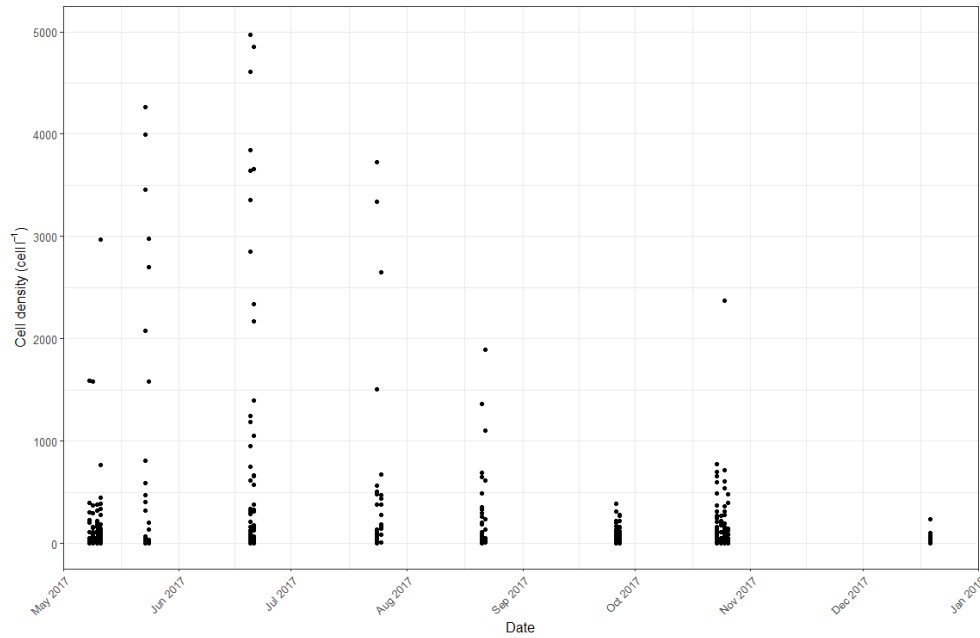


**Figure H2. Diatom cell density in 2017 observed with the LifeWatch Flowcam with a y-axis limited at 5000 cell l⁻¹.**





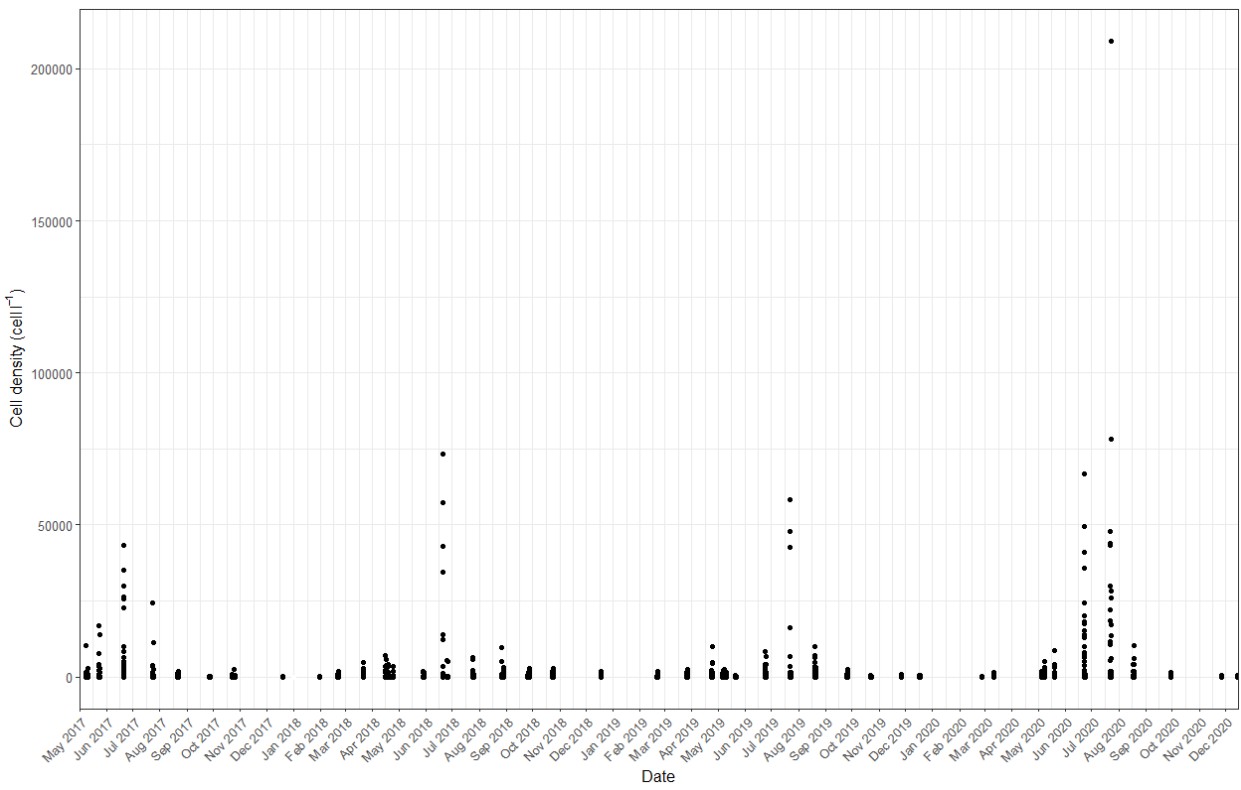

**Figure H3. Diatom cell density from May 2017 to December 2020 observed with the LifeWatch Flowcam.**

**Funding**

This work was supported by the European H2020 Blue-Cloud project [grant number 862409] [V.O., S.P., P.C.]. LS by
Research Foundation - Flanders (FWO) as part of the Belgian contribution to LifeWatch (I002021N-LIFEWATCH).

**Data and code availability**

The results reported in this manuscript have been obtained by using a dedicated working environment: Zoo and Phytoplankton
EOV Products Virtual Laboratory operated by D4Science.org (https://blue-cloud.d4science.org/web/zoo-phytoplankton_eov).
This working environment hosts the data and the software, thus making the process leading to the results repeatable according
to open science practices. The processed data and scripts have also been published in the Blue-Cloud Catalogue
(https://data.d4science.org/ctlg/Zoo-Phytoplankton_EOV/spatiotemporal_analysis_of_plankton_drivers_
in_the_belgian_part_of_the_north_sea_data), as well as deposited in Zenodo (doi.org/10.5281/zenodo.5575442), under a
Creative Commons CC-BY 4.0 license, allowing the use of the data and code under the condition of providing the reference
to the original source:
Otero V., Pint S., Deneudt K., De Rijcke M., Mortelmans J., Schepers L., Cabrera P., Sabbe K., Vyverman W., Vandegehuchte M., Everaert G., 2021: Spatiotemporal analysis of plankton drivers in the Belgian part of the North Sea: Data. Flanders Marine Institute; Ghent University; Department of Biology. https://doi.org/10.5281/zenodo.5575442

**Credit Author statement**

**VO:** Conceptualization, Methodology, Software, Validation, Investigation, Data Curation, Writing - Original Draft, Writing - Review & Editing. **SP:** Software, Validation, Formal analysis, Writing - Original Draft, Writing - Review & Editing, Visualization. **KD:** Writing - Review & Editing. **MDR:** Writing - Review & Editing. **JM:** Investigation, Writing - Review & Editing. **LS:** Writing - Review & Editing. **PC:** Writing - Review & Editing, Project administration. **KS:** Writing - Review & Editing. **WV:** Writing - Review & Editing. **MV:** Writing - Review & Editing. **GE:** Conceptualization, Methodology, Writing - Review & Editing, Supervision.

**Competing interest statement**

The authors declare no competing interest

**Acknowledgements**

The production of this work has been supported by the Blue Cloud working environment via the Blue Cloud (https://blue-cloud.d4science.org) operated by D4Science.org, www.d4science.org (Assante et al., 2019). This work makes use of the LifeWatch data and infrastructure funded by Research Foundation - Flanders (FWO) as part of the Belgian contribution to LifeWatch ESFRI.

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
