# Peer review of "Pronounced seasonal and spatial variability in determinants of phytoplankton biomass dynamics along a near–offshore gradient in the southern North Sea"

_Biogeosciences, 2022_

## Author Comment (AC1)

**Authors' response to the referees' comments**

We appreciated and welcomed the comments from the referees and with this response letter we want to share with the editor our intentions on how we will tackle each of the comments. Based on our replies and intentions we hope to be allowed to go to the second phase of the review process, being the submission of a revised manuscript. The referees' comments were listed, enumerated and answered below.

Referee 1:

1. Limitations are not well discussed.

   We acknowledge that the limitations of the model could have been clearer. They are scattered throughout the manuscript. To address this issue, we will create a separate and well-structured section focusing on the model's limitations.

2. Lack of mathematical formalism.

   Mathematical formulas will be formalized in the revised version of the manuscript.

3. Lack of adequate presentation of the equations and processes that compose each state variable and what is the reason to include them.

   In the original paper all mathematical equations are provided in the appendix, but we understand the need to address them in the main text. We will summarize them and add them to the main text, while adhering to the rules for equations.

   a. Why is there only one specific phytoplankton group included in the model?

      Chlorophyll-a was used as a proxy for marine primary production. Please note that chlorophyll-a data were the results of pigment purification and sample analysis, i.e. taking the cumulative sum of phytoplankton taxa present in the water column. These data have been collected in the scope of the LifeWatch framework, and according to SOP in the labs as defined in the LifeWatch project (Mortelmans et al., 2019). We agree this was a bit obscure in the original version of the manuscript, but intend to indicate and stress this at a few more occasions in a revised version.

   b. Fixed Chl:C:N:P ratios which are used across regions.

      It is a correct observation that we are working with fixed C:Chl ratios in the calibrated models. However, please note that these were selected during the calibration of the NPZD model of each region separately. This means that the Chl:C:N ratios could have any value between the min. max. ranges as defined in Table B1. The limitations of working with the fixed Chl:C:N ratios in the calibrated models have been discussed in the original version. Anugerahanti et al. (2021) state that fixed C:Chl ratios are

commonly used in biogeochemical models, e.g. Diat-HAdCC model (Totterdell, 2019), even with its well-known limitations.

c.      Why is there only one single zooplankton grazer included in the model?

We grouped the most abundant and dominant species of the southern North Sea, therefore we are not using a single zooplankton grazer. The zooplankton densities are dominated by smaller neritic copepods (66%), such as *T. longicornis* and *A. clausi*, together with the appendicularian *O. dioica* (10%; Van Ginderdeuren et al., 2014). This was mentioned in the original version of the manuscript in the second paragraph of the M&M section and the first paragraph of Appendix C. We will rephrase this part to ensure this is clear to all readers.

d.      Fixed sinusoidal surface irradiance (simple Lambert-Beer exponential decay function only applied at 3m depth) and no justification to use it across regions.

The angle of the sunlight and surface was taken into account together with the seasonal variation in light hours. Across the small area that we are working with, the difference in irradiance between the different regions (latitude and longitude) is neglectable. We agree that this forms a limitation to extrapolate the model to other areas. We will specifically mention this in the limitations paragraph of the revised version of the manuscript.

As the used environmental data was collected at 3m depth, we calculated the irradiance at the same depth. In addition, we included the turbidity (Kd) in our calculations. We agree this was a bit hidden in our original version of the manuscript, and will be made more clear in our revision. Lastly, the North Sea is a turbid, restricting sunlight from penetrating deep in the water column.

e.      The model does not include vertical mixing or other types of transport.

The Belgian part of the North Sea is continuously vertically mixed, which will be added in the description of the study area. Adding to that, vertical mixing is inherently included in the in situ observations which have been used to calibrate and run the model.

Horizontal mixing is inherently included in the observations used to model the nutrients. This was briefly mentioned in the paragraph on the analysis of the model in M&M, but we will elaborate further on this topic both in M&M as well as in the discussion section.

4.     Only a very superficial description of the model is provided in the appendix.

A summary of the model was provided in the appendix, as the main ecological principles behind the model are well established (Daewel et al., 2019; Tian et al., 2015; Aarflot et al., 2022) and extensively described in Soetaert and Herman (2009). In the revised manuscript,

we will move part of the model equations to the main text in order to be more transparent on the state variables and interactions.

5.    Validation needs to be better presented. Referee 1 suggests a plot of model predictions against observations.

We appreciate the referee's suggestion and will include such a plot in the appendix of the revised version of the manuscript.

6.    The comparison with other coastal and regional seas models needs to be better discussed. It is unclear how co-limitation is untangled with a simple NPZD model, in what they call "relative contribution".

We decided to use the NPZD model based on the available data that we have in the Belgian part of the North Sea via LifeWatch. Though the ERSEM model is similar in purpose and overcomes some of the limitations in the NPZD model, we focused on using in-situ data collected via LifeWatch. Additionally, the HAMSOM model is focused on hydrodynamics, which is not the purpose of the NPZD model, therefore it is not possible to make a direct comparison. We will add these discussion topics to the revised version of the manuscript.

The term "relative contribution" is an accepted term, which has been used in other research as well, e.g. Deschutter et al. (2017), Everaert et al. (2015), McMahon et al. (2021) and Velthuis et al. (2017). We agree that a clear definition was lacking, hence we will add that in the revised version of the manuscript. We see quantified co-limitation as a synonym for 'relative contribution'.

=> These aspects remain unanswered and present a more interesting research venue than just predicting chlorophyll-a and overinflating the implications of a model with still many reservations.

The limitations of the model will be more transparent and stressed in the revised manuscript. In addition, we want to clarify the goal of the project 'Blue-Cloud' in the scope of which this research was performed. The Blue-Cloud aims at providing an Open Science platform that is freely accessible to everyone enhancing collaborative marine research. The idea of the Blue-Cloud, and the Zoo- and Phytoplankton EOV demonstrator in particular, is to provide a description of the current state of the plankton communities and forecast their evolution, representing valuable information for the modeling, assessment and management of the marine ecosystem. The Blue-Cloud state that it is useful for a variety of communities:

- Researchers can use the model to simulate different scenarios at new scales of observations (e.g. regional/global, seasonal and time series).
- Fisheries advisory organizations can use these plankton products to assess the availability of food resources and its effects on fish stocks.
- Marine policy officers will have the needed support to address  European policy and societal challenges, such as food insecurity, as foreseen under the EU Biodiversity Strategy for 2030.

We will add this context in the revised version of the manuscript.

In addition to just predicting Chlorophyll-a, the model also predicts co-limitation and how much each state variable contributes to the changes in phytoplankton biomass dynamics.

Referee 2:

1. Address some of the applications already for this paper / opportunity to add section to depict implications of their findings instead of listing it as future scope (in term of fishery, HABs, maritime use conflicts/hazards (invasive species)

We believe that this is beyond the scope of the manuscript. The focus of this manuscript was on the model development and quantification of the limiting factors for the plankton dynamics. We feel the manuscript might become too long in case an application is added to the revised version of the manuscript. The follow-up with applications of this model will be taken up in the following years together with the desired documentation.

2. Include additional validation, e.g. remote sensing or in-situ data

We used quality in-situ data of an established research infrastructure, i.e. LifeWatch, to describe the dynamics of phytoplankton in the Belgian North Sea. In addition, before remote sensing data could be used to validate our model results, it should be properly corrected for the different methodology that is used in remote sensing. Again this would add an extra layer of complexity which we feel would not improve the quality of the revised version of the manuscript.

References:

Aarflot, J.M., Hjøllo, S.S., Strand, E., Skogen, M. D.: Transportation and predation control structures the distribution of a key calanoid in the Nordic Seas, Progress in Oceanography, 202, https://doi.org/10.1016/j.pocean.2022.102761, 2022

Anugerahanti P., Kerimoglu O. and Smith S.L.: Enhancing Ocean Biogeochemical Models With Phytoplankton Variable Composition. Front. Mar. Sci. 8:675428, doi: 10.3389/fmars.2021.675428, 2021

Daewel, U., Schrum, C., and Macdonald, J. I.: Towards end-to-end (E2E) modelling in a consistent NPZD-F modelling framework (ECOSMO E2E_v1.0): application to the North Sea and Baltic Sea, Geosci. Model Dev., 12, 1765–1789, https://doi.org/10.5194/gmd-12-1765-2019, 2019.

Deschutter, Y., Everaert, G., De Schamphelaere, K. and De Troch, M.: Relative contribution of multiple stressors on copepod density and diversity dynamics in the Belgian part of the North Sea, Mar. Pollut. Bull., 125(1–2), 350–359, doi:10.1016/j.marpolbul.2017.09.038, 2017.

Everaert, G., De Laender, F., Goethals, P. L. M. and Janssen, C. R.: Relative contribution of persistent organic pollutants to marine phytoplankton biomass dynamics in the North Sea and the Kattegat, Chemosphere, 134, 76–83, doi:10.1016/j.chemosphere.2015.03.084, 2015.

McMahon, K. W., Ambrose, W. G., Reynolds, M. J., Johnson, B. J., Whiting, A., Clough, L. M.: Arctic lagoon and nearshore food webs: Relative contributions of terrestrial organic matter, phytoplankton, and phytobenthos vary with consumer foraging dynamics, Estuarine, Coastal and Shelf Science, 257, https://doi.org/10.1016/j.ecss.2021.107388, 2021.

Mortelmans, J., Deneudt, K., Cattrijsse, A., Beauchard, O., Daveloose, I., Vyverman, W., Vanaverbeke, J., Timmermans, K., 750 Peene, J., Roose, P., Knockaert, M., Chou, L., Sanders, R., Stinchcombe, M., Kimpe, P., Lammens, S., Theetaert, H., Gkritzalis, T., Hernandez, F. and Mees, J.: Nutrient, pigment, suspended matter and turbidity measurements in the Belgian part of the North Sea, Sci. Data, 6(1), 1–8, doi:10.1038/s41597-019-0032-7, 2019.

Tian, R., Chen, C., Qi, J., Ji, R., Beardsley, R.C., Davis, C.: Model study of nutrient and phytoplankton dynamics in the Gulf of Maine: patterns and drivers for seasonal and interannual variability, ICES Journal of Marine Science, Volume 72, Issue 2, January/February, Pages 388–402, https://doi.org/10.1093/icesjms/fsu090, 2015

Totterdell, I. J.: Description and evaluation of the Diat-HadOCC model v1.0: the ocean biogeochemical component of HadGEM2-ES, Geosci. Model Dev., 12, 4497–4549, https://doi.org/10.5194/gmd-12-4497-2019, 2019.

Van Ginderdeuren, K., Van Hoey, G., Vincx, M. and Hostens, K.: The mesozooplankton community of the Belgian shelf (North Sea), J. Sea Res., 85, 48–58, doi:https://doi.org/10.1016/j.seares.2013.10.003, 2014.

Velthuis, M., de Senerpont Domis, L. N., Frenken, T., Stephan, S., Kazanjian, G., Aben, R., Hilt, S., Kosten, S., van Donk, E., and Van de Waal, D. B.: Warming advances top-down control and reduces producer biomass in a freshwater plankton community. Ecosphere 8(1):e01651. 10.1002/ecs2.1651, 2017